# High levels of histones promote whole-genome-duplications and trigger a Swe1[WEE1]-dependent phosphorylation of Cdc28[CDK1]

**Douglas Maya Miles[1†*], Xenia Peñate[2], Trinidad Sanmartín Olmo[3,4], Frederic Jourquin[1], Maria Cruz Muñoz Centeno[2], Manuel Mendoza[3,4], Marie-Noelle Simon[1], Sebastian Chavez[2], Vincent Geli[1†*]**

[1]Marseille Cancer Research Center (CRCM), U1068 Inserm, UMR7258 CNRS, Aix-Marseille Université, Institut Paoli-Calmettes, Equipe Labellisée Ligue, Marseille, France; [2]Instituto de Biomedicina de Sevilla, Hospital Virgen del Rocío-CSIC-Universidad de Sevilla, Sevilla, Spain; [3]Centre for Genomic Regulation, Barcelona Institute of Science and Technology, Barcelona, Spain; [4]Universitat Pompeu Fabra, Barcelona, Spain

**\*For correspondence:**
douglas.maya@cabimer.es (DMM);
vincent.geli@inserm.fr (VG)

[†]These authors contributed equally to this work

**Competing interests:** The authors declare that no competing interests exist.

**Abstract** Whole-genome duplications (WGDs) have played a central role in the evolution of genomes and constitute an important source of genome instability in cancer. Here, we show in *Saccharomyces cerevisiae* that abnormal accumulations of histones are sufficient to induce WGDs. Our results link these WGDs to a reduced incorporation of the histone variant H2A.Z to chromatin. Moreover, we show that high levels of histones promote Swe1[WEE1] stabilisation thereby triggering the phosphorylation and inhibition of Cdc28[CDK1] through a mechanism different of the canonical DNA damage response. Our results link high levels of histones to a specific type of genome instability that is quite frequently observed in cancer and uncovers a new mechanism that might be able to respond to high levels of histones.
DOI: https://doi.org/10.7554/eLife.35337.001

## Introduction

Chromatin replication requires the synthesis and assembly of nucleosomes that wrap around DNA. Each time a cell divides several millions of histones, small basic proteins that conform nucleosomes, are synthesised and incorporated as the replication machinery copies the genome. Higher eukaryotes are unable to survive without histones and possess several alternative pathways to ensure enough histones during replication (*Cook et al., 2011*; *Groth et al., 2005*; *Marzluff et al., 2008*). Cells can also modulate cell cycle progression when histones become limiting to ensure the faithful replication of chromatin and avoid genome instability (*Groth et al., 2007*; *Günesdogan et al., 2014*; *Murillo-Pineda et al., 2014*). Histone excess has also been linked to genome instability and to a wide variety of processes in the cell including DNA repair and life span, which probably explains why all eukaryotes have several redundant pathways that ensure the absence of free histones beyond replication (*Au et al., 2008*; *Castillo et al., 2007*; *Feser et al., 2010*; *Gunjan and Verreault, 2003*; *Singh et al., 2010*; *Takayama et al., 2010*).

Accurate chromosome segregation is essential to prevent genome instability. Eukaryotic cells have different mechanisms or checkpoints able to specifically sense and respond to different types of errors. These checkpoints are able to modulate the length of the cell cycle and give the cells additional time to solve them ([*Hartwell and Weinert, 1989*]. *Saccharomyces cerevisiae* like most

eukaryotic cells has two major checkpoints able to block cells prior to mitosis: the DNA damage response (DDR) (*Ciccia and Elledge, 2010*) and the spindle assembly checkpoint (SAC) (*Musacchio and Salmon, 2007*). Both of them are able to perform this block inhibiting the cleavage and degradation of the kleisin subunit of the cohesin complex Scc1$^{RAD21}$. This inhibition takes place through a stabilisation of Pds1$^{Securin}$ either by phosphorylation (DDR) (*Sanchez et al., 1999*) or by preventing its degradation through the APC (DDR and SAC) (*Agarwal et al., 2003*; *London and Biggins, 2014*). The SAC can additionally respond to lack of tension keeping Shugosin at the pericentromere, which prevents cohesin cleavage through the inhibition of Esp1$^{Separase}$ (*Clift et al., 2009*; *Nerusheva et al., 2014*). The DDR can also block mitosis through the phosphorylation and inhibition of the G2/M cyclin-dependent kinase Cdc28$^{CDK1}$ at Tyr$_{19}$ (15 in humans) that plays a key role during mitosis (*Agarwal and Cohen-Fix, 2002*; *Rahal and Amon, 2008*; *Zhang et al., 2016*).

Besides these well-characterised checkpoints, studies in *S. cerevisiae* have revealed an additional checkpoint able to respond to actin cytoskeleton perturbations called the morphogenesis checkpoint, which delays cell-cycle progression when the actin cytoskeleton is perturbed (*Lew, 2000*; *McMillan et al., 2002*; *Sakchaisri et al., 2004*; *Sia et al., 1998*). This checkpoint is able to stabilise Swe1$^{WEE1}$, a kinase able to promote a phosphorylation on Tyr$_{19}$ of Cdc28$^{CDK1}$ (Tyr$_{15}$ in humans) that inhibits its activity and results in a delay on the metaphase to anaphase transition (*Lianga et al., 2013*). Swe1$^{WEE1}$ is present in lower and higher eukaryotes during an unperturbed cell cycle. This kinase is expressed during replication and degraded before mitosis in a mechanism that involves the action of several kinases that promote Swe1$^{WEE1}$ hyperphosphorylation and trigger its ubiquitination and subsequent destruction by the proteasome (*Howell and Lew, 2012*). Swe1$^{WEE1}$ can also be stabilised upon DNA damage (*Palou et al., 2015*) and in response to certain types of stress (*Chauhan et al., 2015*; *George et al., 2007*; *King et al., 2013*). Interestingly, Swe1$^{WEE1}$ was quite recently shown to be able to physically interact with histone H2B and promote its phosphorylation. This phosphorylation is conserved from yeasts to humans and seems to play an important role in the repression of histone transcription at the end of S-phase (*Mahajan et al., 2012*).

One crucial question that remains unsolved is whether cells are able to respond or sense high levels of histones as they do when they become limiting in order to prevent their undesirable effects on genome stability. To address this question, we have constructed a set of tools to test and analyse the cellular consequences of abnormal accumulations of histones beyond DNA replication in the budding yeast *S. cerevisiae*. Our results show that cells that fail to degrade histones after DNA replication or wild-type cells constitutively exposed to high levels of histones H2A and H2B have defects in chromosome segregation and are able to suffer aberrant cell divisions that result, in some ocassions in viable cells that have experienced a whole-genome-duplication event (WGD). We observe that high levels of histones promote changes in chromatin structure, increase nucleosome occupancy at centromeric and pericentromeric chromatin and decrease the incorporation of the histone variant Htz1$^{H2A.Z}$ to chromatin. This defect in Htz1$^{H2A.Z}$ incorporation is accompanied by a reduction of condensin recruitment to pericentromeric chromatin, a phenotype that could explain why cells exposed to high levels of histones experience WGDs (*Kim et al., 2009*; *Oliveira et al., 2005*; *Woodward et al., 2016*). We also show that Swe1$^{WEE1}$ is stabilised in the presence of high levels of histones and promotes the phosphorylation of Cdc28$^{CDK1}$ through a mechanism that does not require the activation of the DNA damage response. Our results, link for the first time high levels of histones with a cell cycle mark able to delay the cell cycle transition from G2 to mitosis and highlight histone levels as a potential and yet unexplored source of genome instability.

## Results

### Persistent generation of histones promotes WGDs

To determine the effects of histone overexpression beyond replication, we first decided to use a centromeric vector that expresses histones H2A and H2B under the control of a promoter that is not repressed outside of S-phase (*CEN-HTA1-HTB1ΔNEG* referred as CENΔ*NEG*) (*Osley et al., 1986*). Deletion of the *NEG* regulatory site (Δ*NEG*) in the *HTA1-HTB1* cluster interferes with the normal repression of histones H2A and H2B by the HIR complex and leads to their persistent transcription beyond S-phase (*Figure 1a*) (*Eriksson et al., 2012*). This vector was introduced in wild-type cells and in cells that carried a mutation in *RAD53* (*rad53K227A*) or a deletion of *TOM1* (*tom1Δ*), both of

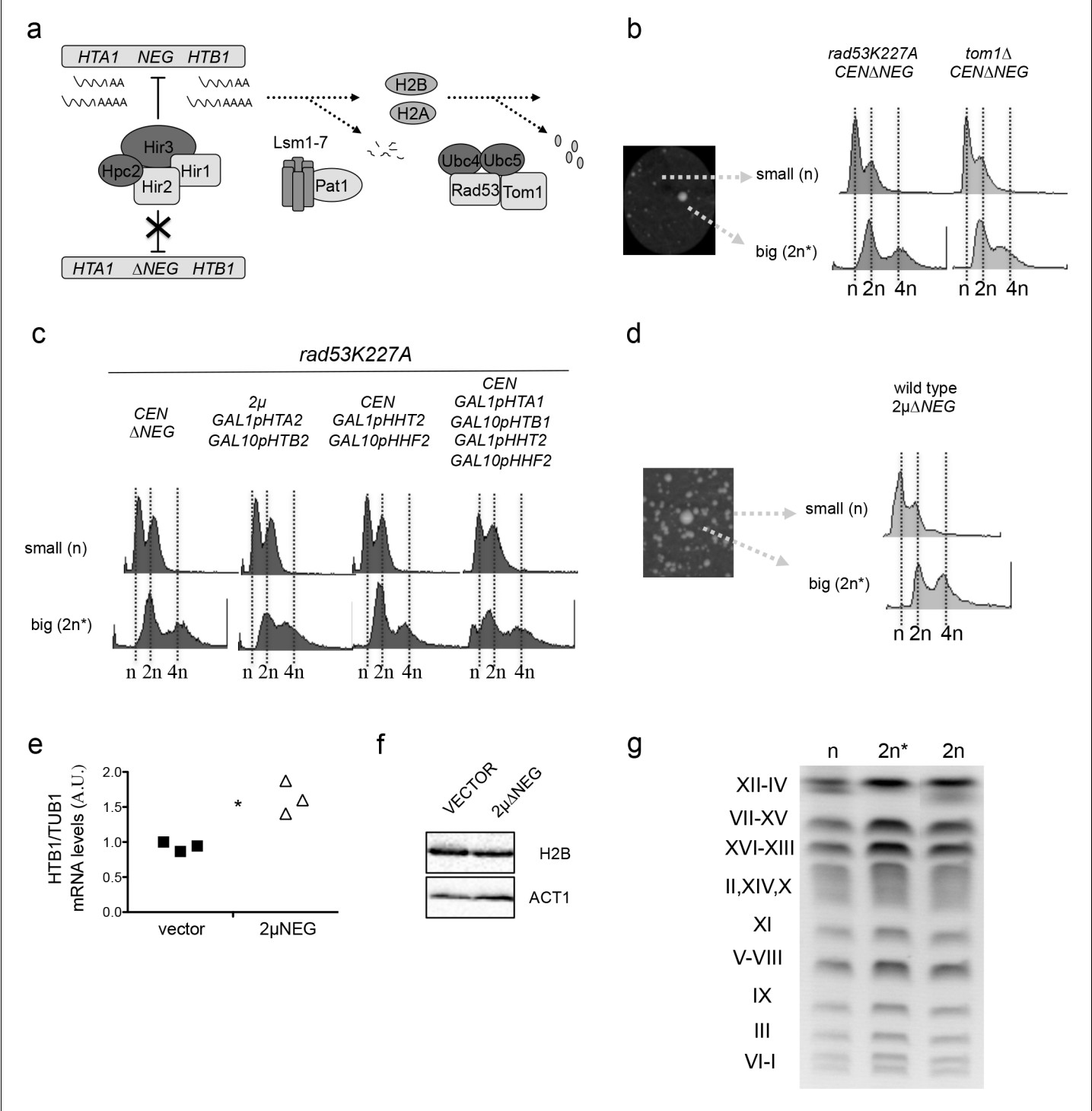

**Figure 1.** Persistent transcription of histones promotes WGDs. (a) Schematic representation of the three pathways that negatively regulate histone levels at a transcriptional (HIR complex), post-transcriptional (Lsm1-7 Pat1 complex), or post-translational (Rad53-Tom1-Ubc4/5 complex) level. (b) Left: small and large colonies observed after transformation of *rad53K227A* or *tom1Δ* cells with the centromeric vector *pHTA1-HTB1ΔNEG* (pCENΔNEG). Right: corresponding FACS profiles (c) FACS profiles from large and small colonies after transformation of *rad53K227A* cells with the indicated vectors. Histone expression is under the control of the *GAL1-GAL10* galactose-inducible divergent promoter. 2μ GAL1pHTA2 GAL10pHTB2 simultaneously expresses histones H2A and H2B, while CEN GAL1pHHT2 GAL10pHHF2 expresses H3 and H4. Simultaneous expression of the four canonical histones is driven by two centromeric vectors. Samples were grown in galactose before and after transformation. (d) FACS profiles from small and big colonies obtained after the transformation of a wild type strain with a high-copy vector that contains the HTA1-HTB1ΔNEG construct (2μΔNEG). (e) H2B (*HTB1*) Q-PCR mRNA levels normalised to *TUB1* mRNA in wild-type cells transformed with the 2μΔNEG or an empty vector (p=0.029; t-test paired samples). *Figure 1 continued on next page*

*Figure 1 continued*

Each point represents an independent experiment (**f**) Representative example of H2B protein levels in wild-type cells transformed with the 2μΔNEG or an empty vector. Act1 was used as a loading control (**g**) Karyotypes from wild-type haploid cells (n), diploid strains (2 n) (obtained by cross) and big colonies obtained after transformation with the 2μΔNEG (2 n*). Karyotypes were analysed by Pulse Field Gel Electrophoresis (PFGE) in five independent clones for each strain.

DOI: https://doi.org/10.7554/eLife.35337.002

The following source data and figure supplement are available for figure 1:

**Source data 1.** Persistent transcription of histones promotes WGDs.

DOI: https://doi.org/10.7554/eLife.35337.004

**Figure supplement 1.** Persistent transcription of histones promotes WGDs.

DOI: https://doi.org/10.7554/eLife.35337.003

them required for the rapid degradation of histone excess (*Gunjan and Verreault, 2003*; *Singh et al., 2009*) (*Figure 1a*). When transformed in *rad53K227A* or *tom1Δ* cells, this construct generated a large number of small colonies that experienced severe growth defects and a small population of large colonies able to grow almost like wild-type cells (*Figure 1b*) (between 2–4 × 10$^{-2}$, raw numbers are in the additional source data file for *Figure 1b*) (*Figure 1—figure supplement 1a*). FACS analysis of these two populations revealed that large colonies had experienced a WGD and became diploid (*Figure 1b*) [detailed information on methods, strain genotype and analyses can be found in the supplemental experimental procedures section]. This effect was not observed in wild-type cells (*Figure 1—figure supplement 1b*) and was not specific for the overexpression of histones H2A and H2B (*Figure 1c*). Growth analysis of small (n) and big colonies (2 n) in the *rad53K227A* mutant confirmed that diploids are able to tolerate better high levels of histones (*Figure 1—figure supplement 1c*). To rule out the possibility that WGDs occur independently of histone levels due to a selection of pre-existing *rad53K227A* diploids endowed with a growth advantage after transformation, we tested if increasing the number of copies of the *HTA1-HTB1ΔNEG* construct would be sufficient to induce WGDs in a wild-type strain. Cells transformed with a high-copy vector that contains the *HTA1-HTB1ΔNEG* construct (2μΔNEG) vector gave rise to a significant number of cells (around 1 × 10$^{-2}$, source data file for *Figure 1b*) that have experienced a WGD (*Figure 1d*). This construct increases by 50% the amount of histone H2B mRNA but does not induce a detectable change in the amounts of total histone H2B protein (*Figure 1e and f*). Diploids were perfectly viable, did not show any major chromosome reorganisation when compared to a haploid or diploid strain obtained by cross (*Figure 1g*) and were able to form triploids when crossed with a strain from the opposite mating type (*Figure 1—figure supplement 1e*). Diploids still had a small growth advantage when compared to haploids in wild-type conditions (*Figure 1—figure supplement 1f*). WGDs were never detected in wild-type cells transformed with an empty vector (more than 2.2 × 10$^2$ colonies analysed by FACS). These results support the fact that WGDs do not take place in the absence of histone deregulation.

## Deregulation of histone levels delays chromosome segregation and promotes aberrant cell divisions in which daughter cells keep all the DNA content

To further understand how WGDs take place, we focused our study on the analysis of wild-type cells transformed with the 2μΔNEG that still remained completely haploid. Time course experiments with these cells revealed that most of them usually turned into a diploid state in 48–96 hr after inoculation in liquid from a petri dish plate (colonies have already grown for 5 days in the plate) (*Figure 2a*). Cell cycle distribution of these haploids transformed with the 2μΔNEG vector in a strain carrying several tags to follow kinetochores (Mtw1-mCHERRY), centrosomes (Spc42-CFP) and tubulin (TUB1-GFP)) revealed a large population of non-dividing cells as well as a significant decrease in the number of cells that have started to separate their chromosomes (*Figure 2b*). Interestingly, in some of these cells we observed that the whole spindle apparatus was able to traverse to the daughter cell before segregation (*Figure 2c*). In order to try to catch the exact moment in which some haploids become diploid, we constructed a strain carrying a tagged nuclear membrane protein and followed cell division using live microscopy in cells transformed with the 2μΔNEG vector or an empty vector. This analysis confirmed a large population of non-dividing cells in the strain transformed with the

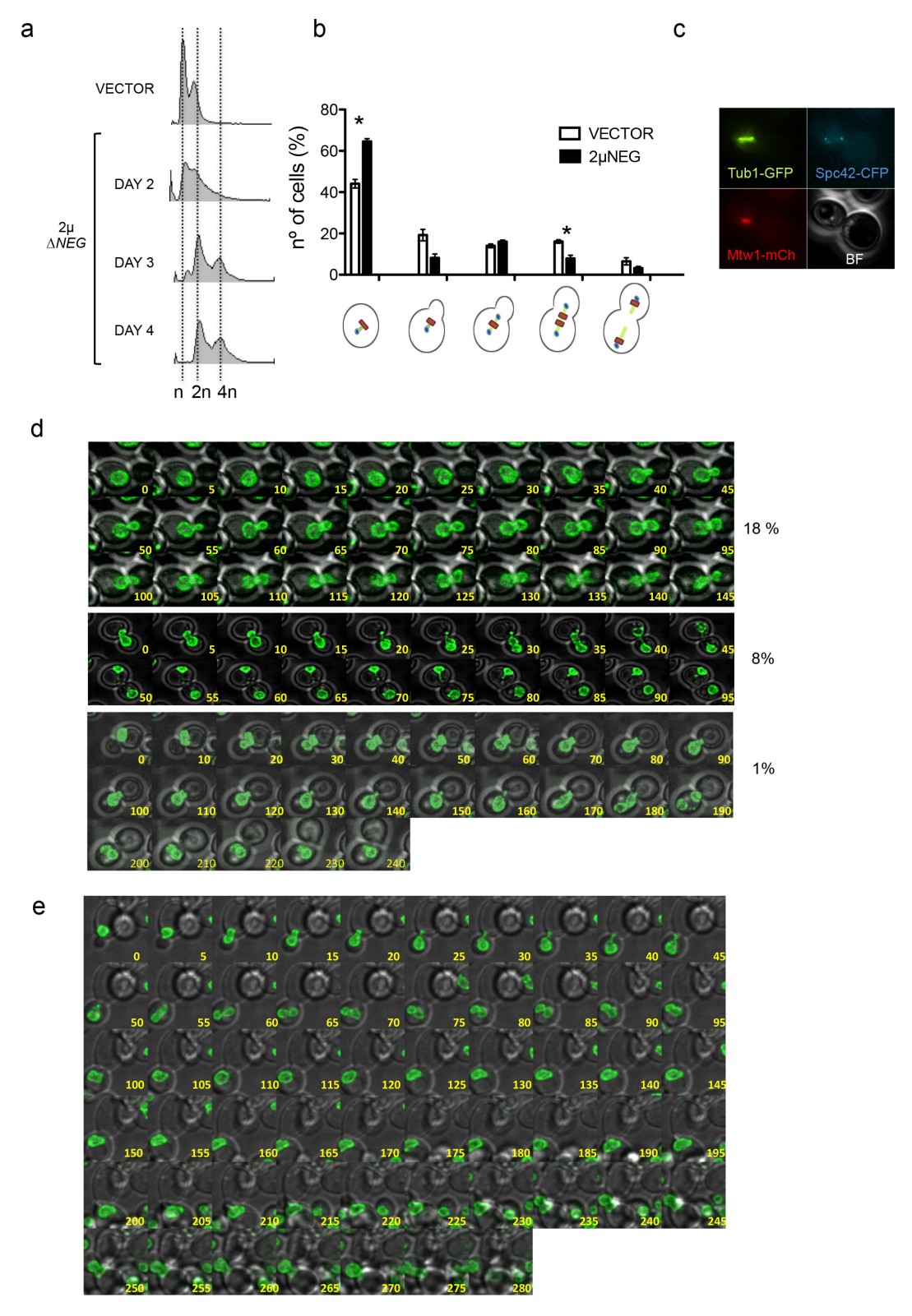

**Figure 2.** Deregulation of histone levels delays chromosome segregation and promotes aberrant cell divisions. (a) Representative FACS profiles of haploid wild type cells after transformation with the 2μΔNEG vector grown during several days. Samples were taken every 24 hr and analysed by FACS to estimate DNA content. (b) Quantification of the different cell cycle stages observed in exponentially growing haploid cells after transformation with the 2μΔNEG or an empty vector (11 Z-sections, 5μ total). The strain (DVY15) carries markers for the kinetochore (Mtw1-mCherry), the spindle pole body
*Figure 2 continued on next page*

*Figure 2 continued*

(Spc42-CFP), and tubulin (Tub1-GFP). Three independent samples were counted for the vector (n = 986) and the 2μΔ*NEG* (n = 1613). Significant p-values are indicated with an asterisk (0.022 and 0.021 from left to right; t-test, paired samples, two tails). (c) Example of a cell in which the whole mitotic machinery can be observed at the daughter cell before mitosis. (d) Live microscopy (5-min interval, 11 Z-sections, 3μ total) of strain DVY12 (Nup49-GFP) after transformation with the 2μΔNEG vector. first panel: Representative example of cells that remain with an undivided nucleus for more than 2 hr (anaphase in control cells usually takes place in less than 20 min). second panel: Representative example of a cell in which the whole nucleus migrates to the daughter before anaphase. third panel: Representative example (10-min interval) of a cell in which both nuclei remain trapped in the daughter cell. Complete movies can be found in *Figure 2d*-videos 1–4. (e) Representative example of a cell in which both nuclei initially trapped in the daughter cell are able to enter a new round of division, and segregate (two nuclei at time 280). Percentages are estimated from the analysis of all movies performed during the first biological replicate (n = 228). Full data from this quantification can be observed in the additional Source Data file.

DOI: https://doi.org/10.7554/eLife.35337.005

The following video and source data are available for figure 2:

**Source data 1.** Deregulation of histone levels delays chromosome segregation and promotes aberrant cell divisions.

DOI: https://doi.org/10.7554/eLife.35337.006

**Figure 2—video 1.** Representative example of a normal anaphase in cells transformed with an empty vector.

DOI: https://doi.org/10.7554/eLife.35337.007

**Figure 2—video 2.** Representative example of cells that remain with an undivided nucleus for more than 2 hr (*Figure 2d* first panel).

DOI: https://doi.org/10.7554/eLife.35337.008

**Figure 2—video 3.** Representative example of a cell in which the whole nucleus migrates to the daughter cell before anaphase (*Figure 2d* second panel).

DOI: https://doi.org/10.7554/eLife.35337.009

**Figure 2—video 4.** Representative example of a cell in which both nuclei remain trapped in the daughter cell (*Figure 2d* third panel).

DOI: https://doi.org/10.7554/eLife.35337.010

2μΔNEG vector, consistent with previous observations (*Morillo-Huesca et al., 2010*), and a significant number of cells (18% vs 3% in the control strain) in which the nucleus remained undivided for several hours (*Figure 2d*, first panel and source data file for *Figure 2d*). This experiment also revealed two specific phenotypes for cells transformed with the 2μΔNEG that were not observed in cells transformed with the empty vector. The first one was a small proportion of cells (8%) in which the nucleus migrated to the daughter cell before mitosis (*Figure 2d*, second panel). The second phenotype involved a combination of the two phenomena previously described and led on some occasions (1%) to an aberrant mitosis in which both nuclei remain trapped in the daughter, generating a binucleated cell (*Figure 2d*, third panel). To confirm that these phenotypes were specific for cells transformed with the 2μΔNEG vector, we performed additional movies comparing cells transformed with this vector or an empty one. These movies confirmed that aberrant mitosis in which the daughter (never observed in the mother) keeps all the genetic material only take place in cells transformed with the 2μΔNEG vector. In one of these events, we were able to follow a second cell division of one of these binucleated cells in which the two nuclear masses were split between the mother and the daughter (*Figure 2e*). We assume that these events may serve as a precursor for diploid cells that are probably being selected among haploids due to their growth advantage.

## Rad53 depletion allows a large and conditional overproduction of histones beyond S-phase in a *lsm1Δ* background

In the experiments described above, we have performed single-cell analysis of cells transformed with a vector that constitutively expresses histones H2A and H2B during the cell cycle. This construct constitutes a useful tool to test the effects of histones per se in genome stability in a mutant-free background. This tool, however, does not allow us to perform molecular studies in wild-type cells exposed to high levels of histones since most of the population becomes fully diploid before large volumes of culture are obtained. Synchronisation experiments in these cells were also problematic. Cells transformed with the 2μΔNEG vector accumulate large amount of cells that do not exit G1 after an alpha factor release or have a very slow progression to S-phase, consistent with previous observations (*Morillo-Huesca et al., 2010*). To overcome these problems and further prove our initial observations, we designed an alternative experimental setting that would theoretically be able to generate large amounts of histones in the cell. We therefore created a strain that combined a deletion of *LSM1*, involved in the posttranscriptional degradation of histone mRNA (*Herrero and Moreno, 2011*) with an Auxin-Inducible Degron of Rad53 (*rad53-AID*) (see *Eriksson et al., 2012*)

and *Figure 1a* for additional information about histone regulation in *S. cerevisiae*). This strategy allowed us to conditionally block at the same time the two major pathways involved in histone degradation at the end of S-phase which is otherwise lethal (*Herrero and Moreno, 2011*).

The *rad53-AID* degron was fully functional as indicated by the low level of Rad53 after one hour of treatment with 500 μM NAA (1-Naphthaleneacetic acid, NAA) (*Figure 3—figure supplement 1a*). *rad53-AID* mutant cells were sensitive to histone overexpression and deletion of *LSM1* was lethal in these cells in the presence of Auxin (*Figure 3—figure supplement 1b and c*). Indeed, 4 hr of Auxin treatment was sufficient to reduce 90% of cell viability (*Figure 3—figure supplement 1d*). This lethality was much higher than the one observed in cells transformed with the 2μΔNEG (*Figure 3—figure supplement 1e*) indicating that this conditional system was much more toxic for the cell.

We next measured histone levels in *rad53-AID LSM1* or *rad53-AID lsm1Δ* cells. To this purpose, we purified core histones (*Jourquin and Géli, 2017*) and measured their amounts relative to a non-specific band in a Comassie Blue-stained acrylamide gel (*Figure 3a*). Quantification of histone levels indicated that Rad53 depletion per se had little or no effect on histone accumulation but did increase histone levels in an *lsm1Δ* background (*Figure 3b*). Western blot analysis with these samples confirmed a significant increase of canonical histones compared to the levels of histone H2A.Z, a histone variant that is not regulated by Lsm1 as canonical histones (*Figure 3c*, *Figure 3—figure supplement 3f*). To address if these changes in chromatin composition were able to affect chromatin structure, we performed a Microccocal Nuclease (MNase) digestion analysis of chromatin extracted from single and double mutants treated or not with NAA (*Figure 3d*). r*ad53-AID-lsm1Δ*-treated cells displayed a clear aberrant nucleosome pattern. While mono and di-nucleosomes were visible at concentrations ranging from 2.5 to 5 mU of Mnase in controls, chromatin from *rad53-AID-lsm1*-treated cells produced a high-molecular-weight smear that suggests that this chromatin is somehow less accesible to Mnase digestion. Overall, our results indicate that the *rad53-AID lsm1Δ* system constitutes a good tool to generate conditional large accumulations of histones in the cell and that high levels of histones are able to promote changes in chromatin structure.

We next sought to investigate if this system phenocopied some of the cell cycle defects observed in cells transformed with the 2μΔNEG vector. Live microscopy and DAPI staining analysis of *rad53-AID-lsm1Δ*-treated cells reveals a large majority of cells that arrest as large budded cells in which the nucleus migrates between the mother and the daughter cell (*Figure 3e* and *Figure 3—video 1* and *2*), a behavior similar to the one observed in the 2μΔNEG cells depicted in the third panel of *Figure 2*. These cells usually died after 6–8 hr of NAA treatment and were unable to enter or skip mitosis. FACS analysis suggested that most of them were able to complete replication (*Figure 3f*). To confirm this observation, we next performed pulse field gel electrophoresis (PFGE) allowing us to differentiate chromosomes in which DNA is fully replicated (chromosomes are able to enter the gel) from chromosomes in which some parts are not (DNA is unable to enter the gel) (*Ide and Kobayashi, 2010*). Cells treated for 4 hr with NAA exhibited a similar pattern to the one observed in cells treated with Nocodazole and clearly different from the one observed in cells treated with Hydroxyurea that leads to the accumulation of cells that have not fully replicated their DNA (*Figure 3g*). To confirm if this system was a useful tool to study the generation of WGDs, we estimated ploidy by FACS in a population of *rad53-AID lsm1Δ* that were platted in rich media plates after 4 hr of treatment with Auxin or DMSO (control). Treatment with Auxin specifically raised a population of cells that had doubled their DNA content (*Figure 3h*). This result reinforces the idea that transient accumulation of high levels of histones are sufficient to promote WGDs in the cell and suggest that the *rad53-AID lsm1Δ* double mutant is a useful tool to study how this process takes place at a molecular level.

## High levels of histones decrease Htz1$^{H2A.Z}$ and condensin incorporation to pericentromeric chromatin

Cse4$^{CENP-A}$ and Htz1$^{H2A.Z}$ are two histone variants that can replace canonical histones H3 and H2A, respectively. Cse4$^{CENP-A}$ lies at the centromere and is essential to recruit the kinetochore (*Stoler et al., 1995*; *Meluh et al., 1998*; *Fachinetti et al., 2013*). Htz1$^{H2A.Z}$ is enriched at the centromere and pericentromeric regions and is required for efficient chromosome segregation (*Albert et al., 2007*; *Krogan et al., 2004*). Cse4$^{CENP-A}$ incorporation outside of the centromere region has been previously linked to WGDs in certain tumours (*Tomonaga et al., 2003*) and to aneuploidy in yeast (*Castillo et al., 2007*; *Collins et al., 2007*). Similarly, high-levels of Htz1$^{H2A.Z}$

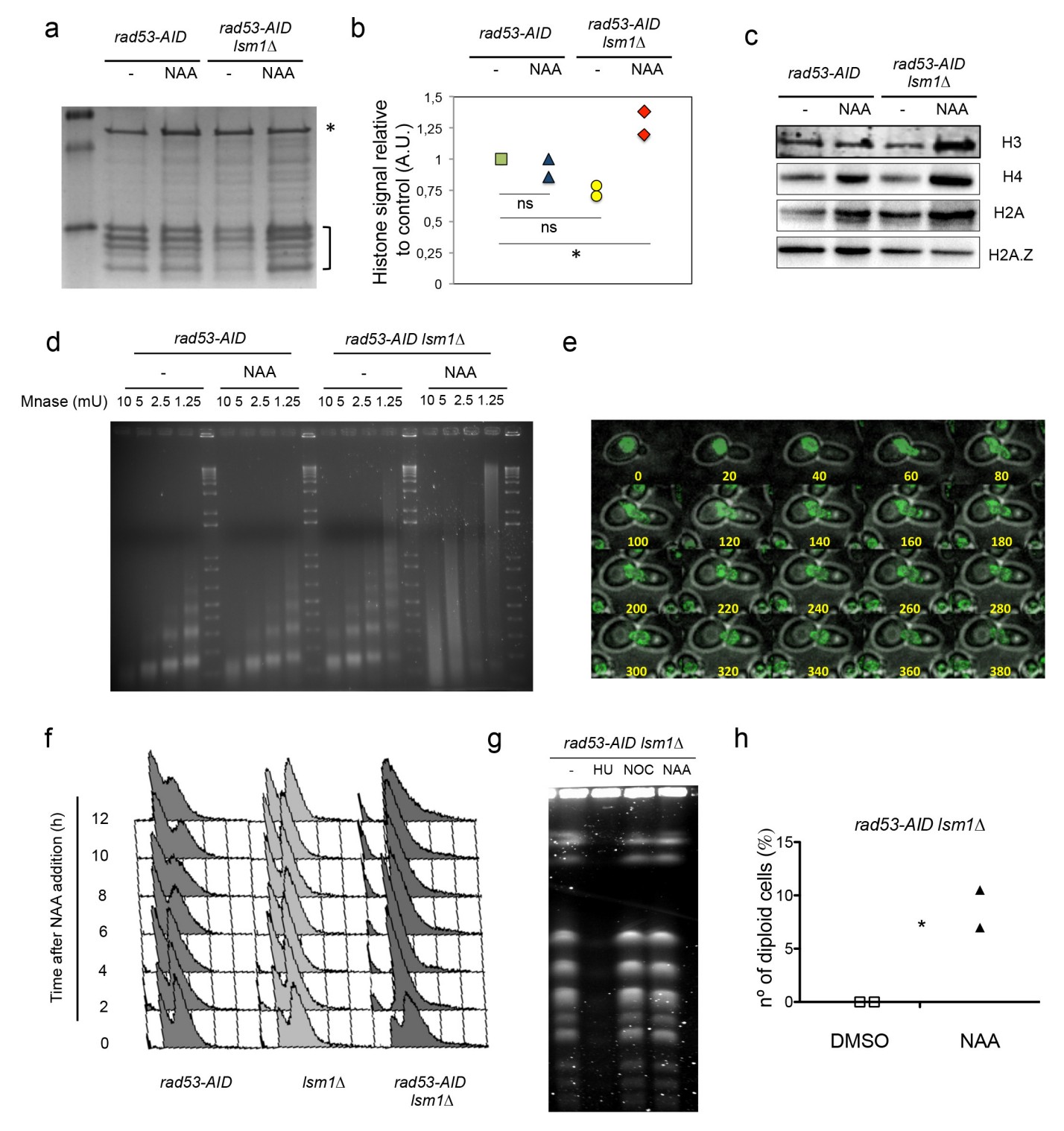

**Figure 3.** Rad53 depletion in *lsm1Δ* cells allows large and conditional overproduction of histones beyond S-phase. (a) Purified core histones from *rad53-AID* and *rad53-AID lsm1Δ* cells that were treated or not 2 hr with NAA and blocked with Nocodazole 2 additional hours. The image represents a Coomassie-Blue-stained gel of histone-purified samples. (b) Quantification of histones in the samples loaded in (a). Histone signal (indicated with a bracket) was normalised using the signal of a non-specific band (indicated with an asterisk). The results were normalised to lane 1. Quantification was done using image J. Two independent experiments were performed. *p=0.026 (t-test, paired, two tails) (c) Western Blots of canonical histones H3, H4 and H2A, and the histone variant H2A.Z. performed with purified core histones samples. Ratio H2A/H4 and H2AZ/H4 are shown in *Figure 3—figure supplement 3f*. (d) Micrococcal nuclease (Mnase) analysis of chromatin extracted from *rad53-AID* and *rad53-AID lsm1Δ* cells treated as described in (a). *Figure 3 continued on next page*

*Figure 3 continued*

(e) Live microscopy (20-min interval, 14 Z-sections, 2,89μ total) in strain DVY36 (*rad53-AID lsm1Δ NUP49-GFP*) after 90 min of Auxin treatment (Auxin was maintained in the media). (f) Cell cycle distribution of *rad53-AID, lsm1Δ and rad53-AID lsm1Δ* cells after treatment with 500 μM NAA. (g) Pulse field gel electrophoresis of *rad53-AID lsm1Δ* cells that were treated with 200 mM Hydroxyurea (HU), 10 μg/ml Nocodazole (NOC), or 500 μM NAA (NAA). Control asynchronous growing cells were not treated (UT). HU-treated cells are blocked in S-phase preventing chromosomes from entering the gel. Nocodazole allows full replication but blocks mitosis. (h) Percentage of *rad53-AID lsm1Δ* cells that have experienced or not a WGD after a 4 hr treatment with 500 μM NAA. The DNA content of all survivors was analysed by FACS. two independent experiments (n = 100 or 75) p-value=0.038 (t-test, two tails, unequal variance).

DOI: https://doi.org/10.7554/eLife.35337.011

The following video, source data, and figure supplement are available for figure 3:

**Source data 1.** Rad53 depletion in lsm1Δ cells allows large and conditional overproduction of histones beyond S-phase.

DOI: https://doi.org/10.7554/eLife.35337.013

**Figure supplement 1.** Rad53 depletion in lsm1Δ cells allows large and conditional overproduction of histones beyond S-phase.

DOI: https://doi.org/10.7554/eLife.35337.012

**Figure 3—video 1.** Videos depicting several examples of the normal behavior of *rad53-aid lsm1Δ NUP49-GFP* cells after a pre-treatment of 90 min with NAA.

DOI: https://doi.org/10.7554/eLife.35337.014

**Figure 3—video 2.** Videos depicting several examples of the normal behavior of rad53-aid lsm1Δ NUP49-GFP cells after a pre-treatment of 90 min with NAA.

DOI: https://doi.org/10.7554/eLife.35337.015

incorporation at the pericentromeric region in yeast also affected ploidy (*Chambers et al., 2012*). Since histone overexpression is able to promote changes in centromeric chromatin (*Au et al., 2008*; *Castillo et al., 2007*; *Salzler et al., 2009*; *Takayama et al., 2010*) we checked if high levels of histones were able to promote WGDs by changing the stoichiometry of canonical histones versus non-canonical histones at centromeric chromatin. Chromatin immunoprecipitation (ChIP) of histones (H3, H4, H2B and H2A) and histone variants (Cse4-Myc$^{CENP-A}$ and Htz1$^{H2A.Z}$) were carried out in *rad53-AID* and *rad53-AID lsm1Δ* mutants that were treated or not with Auxin (*Figures 4* and *5*). These experiments were carried out in the presence of Nocodazole to confirm that all the changes observed are related to histone accumulation and not to changes in the composition of centromeric chromatin related to the cell cycle.

Analysis of several regions of chromosomes IV and XII revealed no changes in the amount of Cse4$^{CENP-A}$ at centromeres (*Figure 4a*, left panel). As expected, incorporation of Cse4 outside of the centromeres was low and differences in the Cse4 at these regions were not significant. Canonical histones in contrast did increase their occupancy along several regions and were highly enriched at centromeres (*Figure 4a*, middle and right panels). To check if kinetochores were still able to attach to centromeres, we measured by ChIP the recruitment of Mtw1, an essential component of the MIND complex that binds centromeric DNA through the association with inner and outer kinetochore elements (*Hornung et al., 2014*) in asynchronous growing cells (*Figure 4b*) and in cells synchronised with Nocodazole (*Figure 4c*). Mtw1 recruitment to centromeres IV and XII was similar and did not significantly increased in other regions of the genome. To confirm that kinetochores remained bound to the spindle, we studied the behaviour of chromosomes along the spindle axis in *rad53-AID lsm1Δ* arrested cells using a strain with tags to follow kinetochores (Mtw1-mCHERRY) and the spindle axis (TUB1-GFP) (*Liu et al., 2008*). As shown in *Figure 4d*, most of the arrested cells displayed one single mass of chromosomes in the daughter or the mother cell that always remained in line with the spindle axis. We concluded that histone overexpression is able to increase histone H3 and H4 occupancy at centromeres and several other regions but this incorporation does not seem to affect CENP-A recruitment, the attachment of kinetochores to chromosomes, or the attachment of both to the spindle.

We next analysed the recruitment of Htz1$^{H2A.Z}$ to chromatin (*Figure 5a*). We observed a clear decrease in the incorporation of Htz1$^{H2A.Z}$ to pericentromeric chromatin in agreement with the global decrease of the H2AZ/H4 ratio shown in *Figure 3c*. This decrease was opposite to what was observed for histones H2B and H2A and indicated that Htz1$^{H2A.Z}$ molecules were being replaced by the canonical histone H2A in the *rad53-AID lsm1Δ* mutant treated with Auxin.

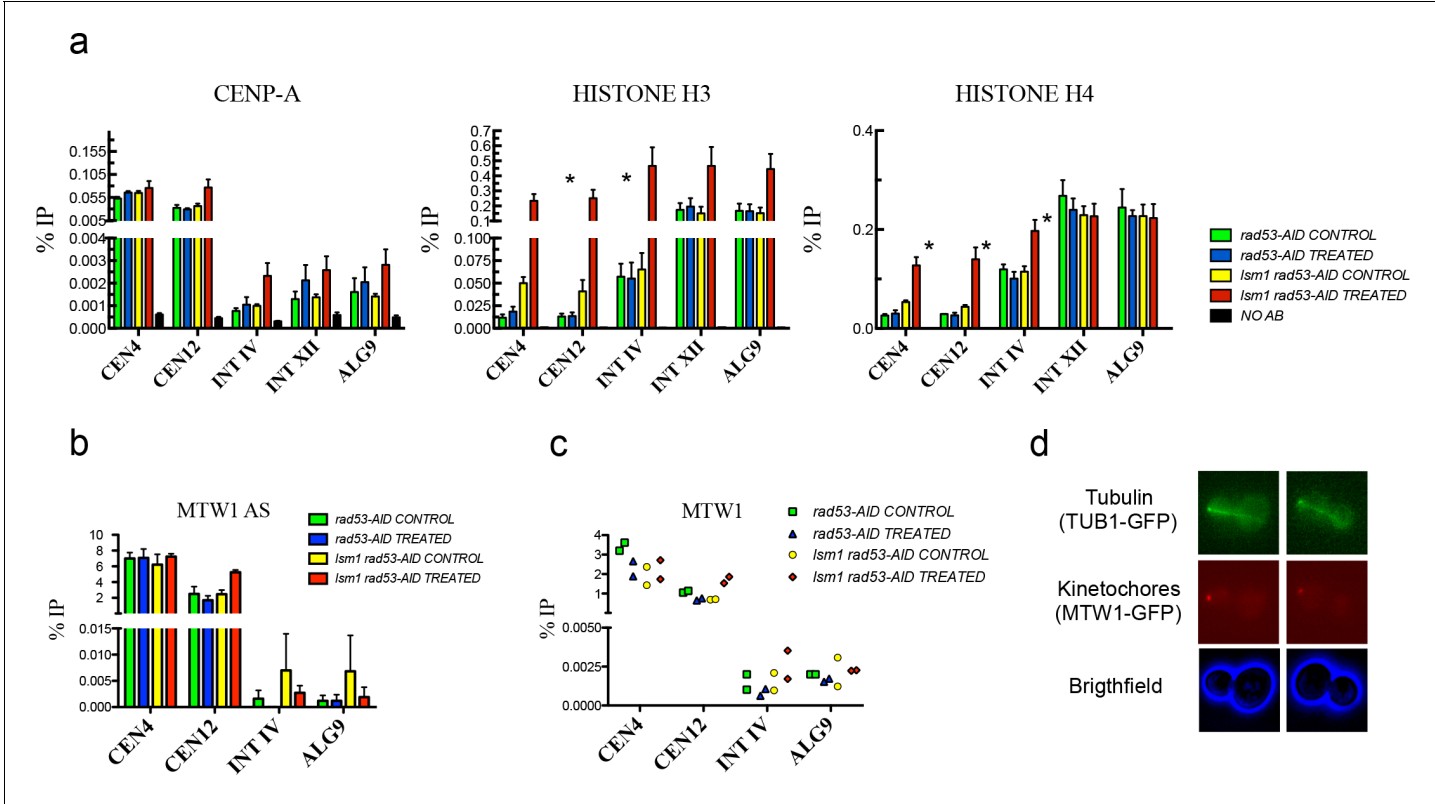

**Figure 4.** High levels of histones do not affect CENP-A recruitment to centromeres or the attachment of chromosomes to the spindle axis. (a) Relative levels of histones H3, H4, Cse4 (CENP-A) measured by ChIP-Q-PCR at the indicated loci in *rad53-AID* (DVY22) and *rad53-AID lsm1Δ* (DVY23) cells. Samples were obtained from cultures treated (or not) with Auxin for 2 hr and incubated 2 additional hours with Nocodazole. CEN4 and CEN12 correspond to the centromeric region of Chromosomes IV and XII. INT IV and INT XII to intergenic regions in the same chromosomes. ALG9 corresponds to a small amplicon in the coding region of ALG9. p-Values were obtained from a Student T-test (paired samples, two tails) that compares *rad53-AID*-untreated cells versus *rad53-AID-lsm1Δ*-treated cells *p<0.05. (b) MTW1 chromatin association (ChIP-qPCR) obtained in two independent experiments in which asynchronous growing *rad53-AID* and *rad53-AID lsm1Δ* cells were treated or not with NAA for 4 hr (c) Same as in b but using samples that were synchronised with NAA and Nocodazole as in (a). (d) Two representative examples of the normal distribution of kinetochores (MTW1-mCherry) along the spindle axis (Tub1-GFP) in *rad53-AID lsm1Δ* cells after 4 hr of Auxin treatment.

DOI: https://doi.org/10.7554/eLife.35337.016

The following source data is available for figure 4:

**Source data 1.** High levels of histones do not affect CENP-A recruitment to centromeres or the attachment of chromosomes to the spindle axis.
DOI: https://doi.org/10.7554/eLife.35337.017

Interestingly, although wild-type cells transformed with the 2μΔNEG vector do not seem to increase the protein levels of H2B (see above *Figure 1f*), the level of Htz1[H2A.Z] was decreased in these cells (*Figure 5b and c*). To investigate if changes in Htz1[H2A.Z] levels were directly related to WGDs, we decided to interfere with the incorporation of this histone variant in both, the 2μΔNEG and the *rad53-AID lsm1Δ* systems. Htz1[H2A.Z] overexpression using a high copy vector largely increased the levels of Htz1[H2A.Z] observed in cells transformed with the 2μΔNEG vector (*Figure 5d*) and was able to decrease the amount of WGDs observed in these cells (*Figure 5e*). Diploid cells were not affected by Htz1[H2A.Z] overexpression discarding an indirect effect due to a counterselection (*Figure 5—figure supplement 1a*). Consistent with the result of *Figure 5e*, deletion of *SWR1*, which decreases Htz1[H2A.Z] incorporation to chromatin (*Mizuguchi et al., 2004*) had an opposite effect to the one conferred by Htz1[H2A.Z] overexpression and increased the formation of WGDs in cells transformed with the 2μΔNEG (*Figure 5f*). Overexpression of histone Htz1[H2A.Z] in *rad53-AID lsm1Δ* cells also increased the amounts of total histone Htz1[H2A.Z] (*Figure 5—figure supplement 5b*) but was unable to supress the defect in the incorporation of this histone variant to chromatin, the G2/M arrest or the lethality after NAA treatment (*Figure 5—figure supplement 1c, d ane e*).

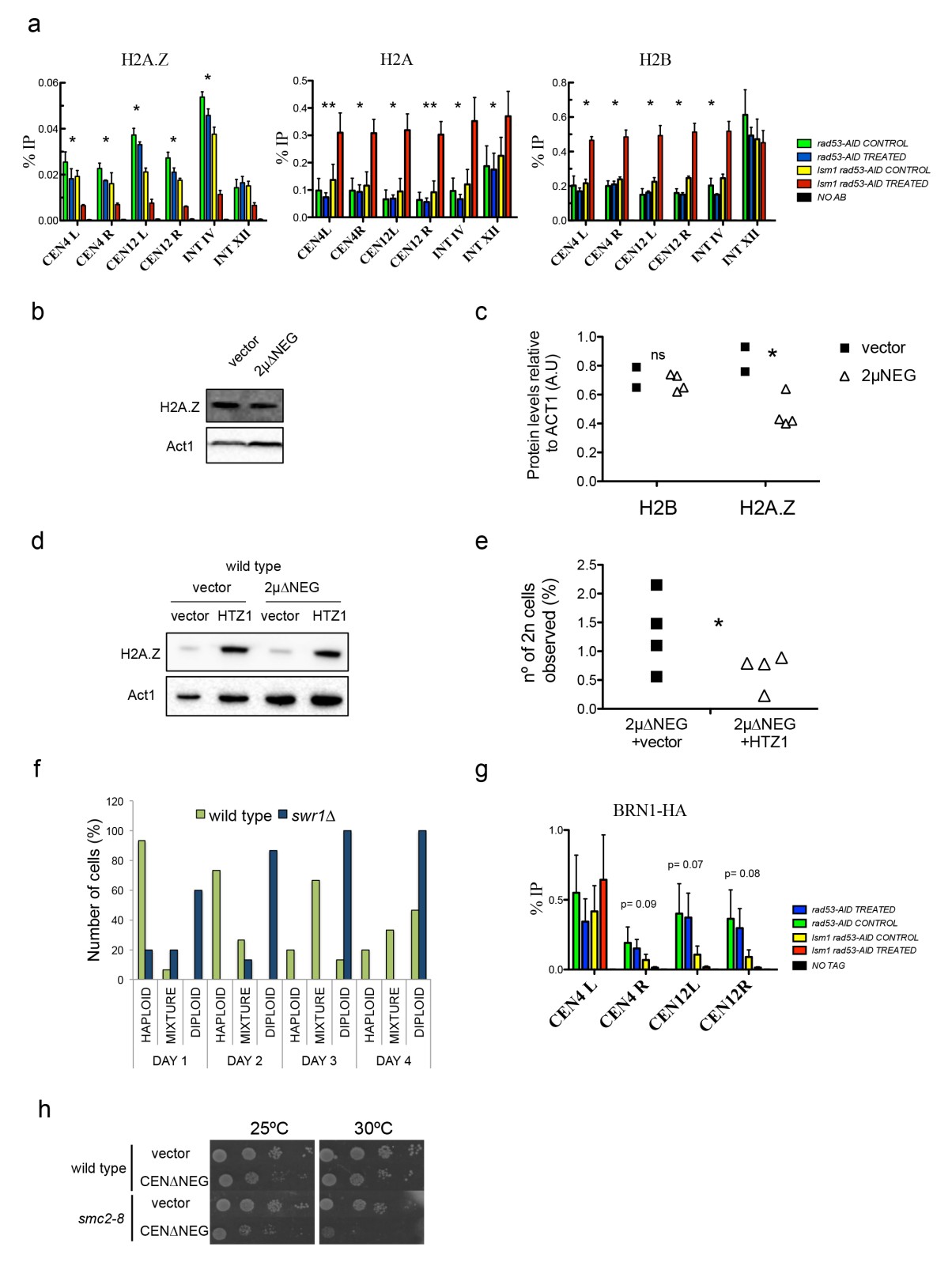

**Figure 5.** High levels of histones decrease Htz1[H2AZ] and condensin incorporation into pericentromeric chromatin. (**a**) Relative levels of histones H2B, H2A and H2A.Z at the indicated loci measured by ChIP-Q-PCR in *rad53-AID* (DVY22) and *rad53-AID lsm1Δ* (DVY23) cells. Samples were obtained from cultures treated (or not) with Auxin for 2 hr and incubated 2 additional hours with Nocodazole. This synchronisation was performed to avoid cell-cycle-related differences. CEN4 (or 12) L (left) and R (right) correspond to the pericentromeric regions of centromeres 4 and 12. p-Values were obtained from

*Figure 5 continued on next page*

*Figure 5 continued*

a Student T-test (paired samples, two tails) that compares *rad53-AID*-untreated cells versus *rad53-AID-lsm1Δ*-treated cells *p<0.05; **p<0.01 (b) Representative example of H2A.Z protein levels obtained in wild-type cells transformed with an empty vector or the 2μΔNEG vector (c) Quantification of protein levels of H2B and H2A.Z relative to Act1. p-Value for H2B differences (t-test, paired, one tail) 0.36; p-value for H2A.Z=0.033 (d) Representative example of H2A.Z protein levels obtained in wild-type cells transformed with either an empty vector or the 2μΔNEG vector. These cells carry a second empty vector (prs425) or the multicopy vector prs425-HTZ1 driving the overexpression of HTZ1 (H2A.Z) (e) WGD events observed in cells transformed with the indicated plasmids. WGDs were estimated as in *Figure 1* (described also in the Supplemental Experimental Procedures). p=0.045 (t-test, paired, one tail) four independent experiments (f) Percentage of cells with a haploid, mixed, or diploid DNA content during a 4 days (D1–D4) time-course in wild type or *swr1Δ* cells after transformation with the *2μΔNEG* vector. Percentages were obtained from the analysis of 15 clones for each background. (g) Relative levels of Brn1 associated to chromatin measured by ChIP-Q-PCR. Experiments and statistical tests were carried out as in (a) in strains DVY32 and DVY33 expressing HA-tagged Brn1. This experiment was performed five times with five independent biological replicates. (h) Plate growth assay of a wild-type strain and a condensin thermosensitive mutant (*smc2-8*) transformed with an empty vector or the centromeric version of the *ΔNEG* vector (CENΔNEG).

DOI: https://doi.org/10.7554/eLife.35337.018

The following source data and figure supplement are available for figure 5:

**Source data 1.** High levels of histones decrease Htz1H2A.Z and condensin incorporation into pericentromeric chromatin.

DOI: https://doi.org/10.7554/eLife.35337.020

**Figure supplement 1.** High levels of histones decrease Htz1H2A.Z and condensin incorporation into pericentromeric chromatin.

DOI: https://doi.org/10.7554/eLife.35337.019

In summary, our results with the 2μΔNEG vector indicate that Htz1$^{H2A.Z}$ relative levels are important in the generation of WGDs in cells exposed to a persistent transcription of histones H2A and H2B. In contrast, overexpression of this histone variant is unable to suppress the defects observed in *rad53-AID-lsm1Δ*-treated cells and suggests that this double mutant might have additional effects on chromatin structure that interfere with Htz1$^{H2A.Z}$ deposition, cell cycle, and cell viability.

Excessive incorporation of Htz1$^{H2A.Z}$ to pericentromeric chromatin has been previously linked to WGDs (*Chambers et al., 2012*). While the opposite has been linked to chromosome segregation defects, it has never been associated to WGDs. In *S. pombe* Htz1$^{H2A.Z}$ is required to stabilise the recruitment of condensin to pericentromeres, a complex able to embrace each of the daughter DNA strands and required to allow their proper segregation (*Cuylen and Haering, 2011*; *Kim et al., 2009*). Depletion of the CAP-H kleisin subunit (Brn1) of Condensin II in drosophila S2 cells has been shown to lead to chromosome segregation defects and produce a small number of abortive mitosis in which all the genetic material is retained in one cell (*Oliveira et al., 2005*). Mutations in the condensin II subunit Caph2 in mice have also been recently involved in the generation of cells with an increased ploidy content (*Woodward et al., 2016*). Based on these published results, we tested if Htz1$^{H2A.Z}$ defective recruitment observed in the *rad53-AID lsm1Δ* double mutant would affect condensin recruitment to pericentromeric chromatin (*Figure 5g*). Recruitment of Brn1 to pericentromeric chromatin decreased in three out of the four pericentromeres examined in five independent experiments suggesting that Htz1$^{H2A.Z}$ is important for condensin recruitment to pericentromeric regions in *S. cerevisiae.* We further tested the effects of histone overexpression in a strain that carries a thermosensitive version of the condensin subunit SMC2 (*smc2-8*). The *smc2-8* mutant, was extremely sensitive to histone overexpression (*Figure 5h*) further linking histone levels to condensin function. Cell cycle kinetics to measure WGDs in these cells was not possible due to the extreme sensitivity of this mutant to histone overexpression.

## Accumulation of histones triggers a DNA damage-independent phosphorylation of Cdc28$^{CDK1}$

One of the initial goals of this work was to evaluate if cells have mechanisms able to modulate cell cycle progression in the presence of abnormal levels of histones as they do when histones become limiting during replication. This question is particularly relevant since persistent activation of the DDR, the SAC and Swe1 overexpression have all been linked to WGDs (*Davoli et al., 2010*; *Sotillo et al., 2007*; *Kawasaki et al., 2003*). We therefore investigated if high levels of histones were able to promote the activation of the DDR, the SAC or affect Swe1 stability using our two systems (*rad53-AID lsm1Δ and* 2μΔNEG) able to deregulate histone levels during the cell cycle.

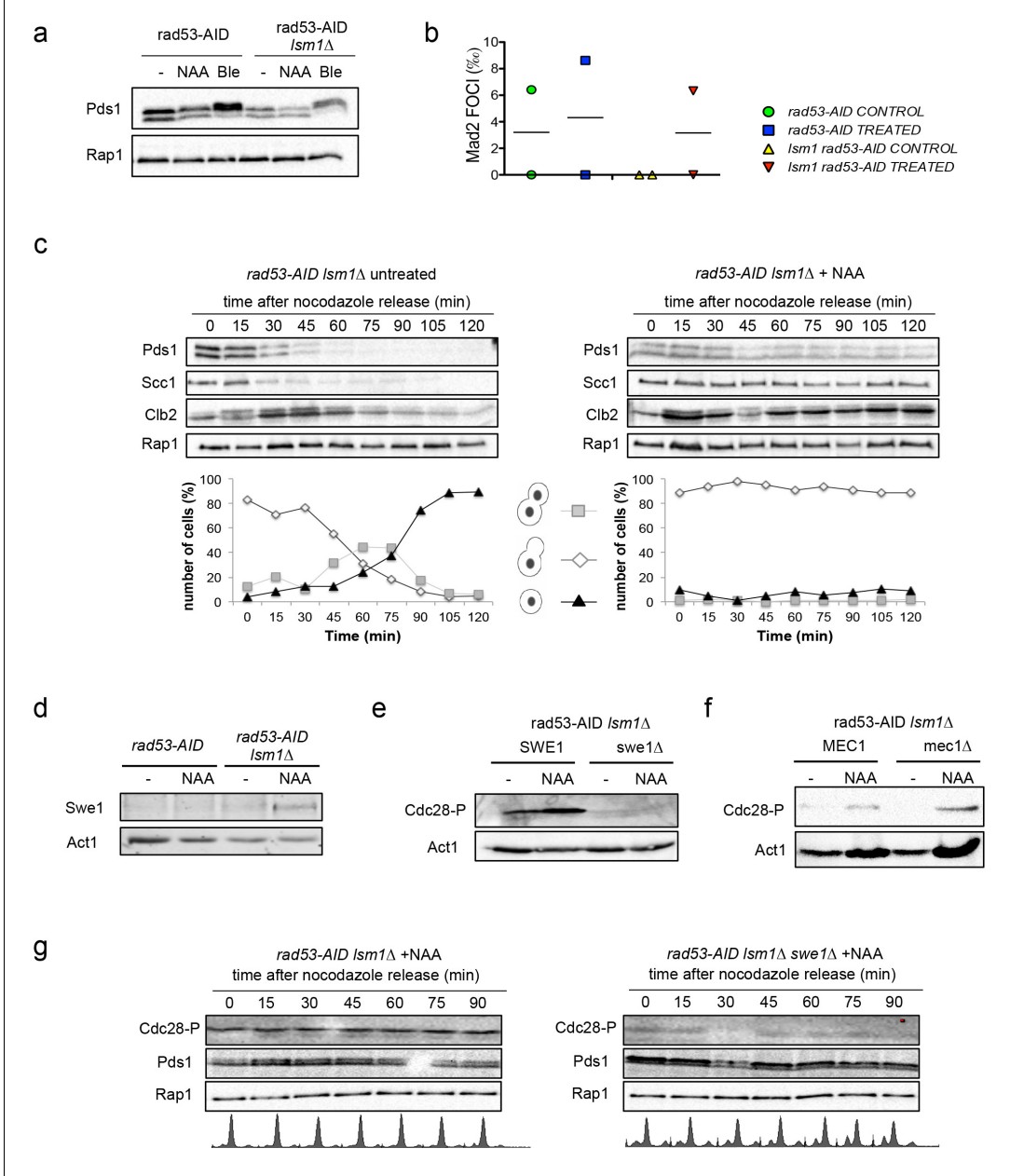

**Figure 6.** Histone accumulation triggers a Swe1-dependent Cdc28$^{CDK1}$ phosphorylation. (**a**) Pds1 phosphorylation (Pds1-HA) in *rad53-AID* and *rad53-AID lsm1Δ* cells either after 4 hr treatment with NAA or bleomycin (Ble, 15 µg/ml). Pds1 appears as a doublet. Its phosphorylation can be visualised as a shifted band after bleomycin treatment. (**b**) Average number of Mad2 foci (Mad2-GFP) that co-localise with kinetochores (Mtw1-mCHERRY) in *rad53-AID* and *rad53-AID lsm1Δ* cells after 4 hr of treatment with either DMSO or Auxin. Two independent experiments were performed in which at least 100 cells were counted. (**c**) Cell-cycle progression from G2/M to the next G1 in *rad53-AID lsm1Δ* cells after treatment or not with Auxin. Exponentially growing cells were treated (or not) with Auxin for 2 hr and incubated for 2 additional hours with Nocodazole. Cells were then washed to eliminate Nocodazole and released into media containing Auxin and alpha-factor to arrest cells able to enter mitosis the next G1. This experiment was performed in cells carrying either a *PDS1-HA* tag or a *SCC1-MYC* tag. The presence of Clb2 and Rap1 was monitored with anti-Clb2 and anti-Rap1 antibodies. Rap1 was used in both as a loading control. A representative full example of the experiment for each strain is shown in the *Figure 6—figure supplement 1a and b*. This data file also includes the results obtained with the single mutant that are not shown in the this figure. Graphs below each panel represent the progression of cells to the next G1. These percentages were estimated from fixed cells that were analysed using DAPI staining. (**d**) Swe1 protein levels in *rad53-AID* and *rad53-AID lsm1Δ* cells treated as indicated. (**e and f**) Cdc28$^{CDK1}$ phosphorylation levels in the indicated strains. Cdc28$^{CDK1}$ phosphorylation is monitored with phospho-cdc2 (Tyr15) antibodies. Cells were treated as indicated (**g**) Cell-cycle progression from G2/M to the next G1 in *rad53-AID lsm1Δ* and *rad53-AID lsm1Δ swe1Δ* similar to the one described in (6d), using the strain that expresses Pds1-HA. The results obtained

*Figure 6 continued on next page*

*Figure 6 continued*

with the single mutant *rad53-AID* are shown in the *Figure 6—figure supplement 1c*. All the experiments observed in *Figure 6* were performed twice and gave similar results.

DOI: https://doi.org/10.7554/eLife.35337.021

The following source data and figure supplement are available for figure 6:

**Source data 1.** Histone accumulation triggers a Swe1-996 dependent Cdc28CDK1 phosphorylation.

DOI: https://doi.org/10.7554/eLife.35337.023

**Figure supplement 1.** (a and b) Representative full examples of the experiments described in *Figure 6d*.

DOI: https://doi.org/10.7554/eLife.35337.022

Auxin treatment in *rad53-AID lsm1Δ* cells did not promote Pds1$^{Securin}$ phosphorylation (*Figure 6a*), a marker for DDR activation, or enhanced the formation of Mad2 foci at kinetochores, a marker used to detect SAC activation (*Figure 6b*). To further confirm or discard the activation of the SAC, we decided to follow Pds1$^{SECURIN}$ degradation during a G2/M release in *rad53-AID and rad53-AID lsm1Δ* cells treated or not with NAA before the release. This experiment revealed a slight stabilisation of Pds1$^{SECURIN}$ in *rad53-AID-lsm1Δ*-treated cells, higher levels of Scc1$^{Rad21}$, the main substrate of Pds1, and Clb2 (*Figure 6c* and *Figure 6—figure supplement 1a and b*). Clb2 started to degrade but reaccumulated later suggesting that the activity of APC-Cdh1 is impaired in this mutant. We next looked at Swe1$^{WEE1}$ levels and observed that this kinase was stabilised in a hypophosphorylated form in *rad53-lsm1Δ*-treated cells (*Figure 6d*). This stabilisation led to a higher level of Cdc28$^{CDK1}$ phosphorylation that depend on Swe1$^{WEE1}$ (*Figure 6e*) and was independent of the DDR, since MEC1 deletion did not abolish it (*Figure 6f*). SWE1 deletion was able to suppress Cdc28$^{CDK1}$ phosphorylation but unable to suppress Pds1 stabilisation suggesting that the mechanism by which Pds1 is stabilised does not depend on Cdc28$^{CDK1}$ activity (*Figure 6g* and *Figure 6—figure supplement 1c*). The results obtained with the *rad53-AID lsm1Δ* system suggest that histone accumulation can stabilise Pds1 and delay cohesion degradation, and that it promotes a Swe1-dependent phosphorylation of Cdc28$^{CDK1}$. The absence of Pds1$^{SECURIN}$ phosphorylation added to the fact that its stabilisation does not depend on Cdc28$^{CDK1}$ phosphorylation favor the idea that Pds1 stabilisation could be due to an activation of the SAC.

We then tested checkpoint activation in cells transformed with the 2μΔNEG vector. Cells transformed with this vector did not trigger Rad53$^{CHK2}$ phosphorylation or enhanced the formation of Ddc2$^{ATRIP}$ foci, two common markers of the DNA damage checkpoint (*Figure 7a and b*). The fact that Mad2 foci were slightly increased (*Figure 7c*) suggest that the SAC could be activated at least in some cells when histones are overexpressed. We failed to detect Cdc28$^{CDK1}$ phosphorylation in asynchronous wild-type cells transformed with an empty vector or the 2μΔNEG construct (data not shown). To confirm if Cdc28$^{CDK1}$ phosphorylation was related to high levels of histones, we compared the phosphorylation kinetics of Cdc28$^{CDK1}$ during one complete cell cycle (from G1 to the next G1) in wild type and *rad53K227A* cells in which histones H2A and H2B were overexpressed from a galactose inducible promoter (*Figure 7d*). Cdc28$^{CDK1}$ phosphorylation in wild-type cells with no histone overexpression was normally enhanced as cells entered replication and decreased when they reached the G2/M transition (*Figure 7d* upper left panel). This phosphorylation was only modestly affected in wild-type cells overexpressing histones (*Figure 7d* upper right panel) but was maintained for a longer time when histones were expressed in the *rad53K227A* strain (*Figure 7d* compare lower left and right panel). Cells exposed to high levels of histones seem to be able to promote a Cdc28$^{CDK1}$ phosphorylation, at least when Rad53 activity is impaired. This result suggest that the Cdc28$^{CDK1}$ phosphorylation observed in *rad53-AID-lsm1Δ*-treated cells is directly related to the presence of high levels of histones.

To address which is the contribution of these proteins to both, the arrest observed in *rad53-AID-lsm1Δ*-treated cells and to the generation of WGDs in 2μΔNEG transformed cells, we finally performed cell cycle kinetics in *rad53-AID lsm1Δ* cells and wild-type cells transformed with the 2μΔNEG vector in cells lacking Pds1$^{Securin}$, Mad2$^{MAD2}$ and/or Swe1$^{WEE1}$. *PDS1* deletion had only a slight effect on the cell cycle arrest observed in *rad53-AID-lsm1Δ*-treated cells and did not significantly change the kinetics of diploidisation observed in cells transformed with the 2μΔNEG vector (*Figure 8—figure supplement 1a–c*). *MAD2* and *SWE1* deletions were able to accelerate the generation of WGDs in 2μΔNEG transformed cells (*Figure 8a*) and did not change the ploidy content of cells transformed

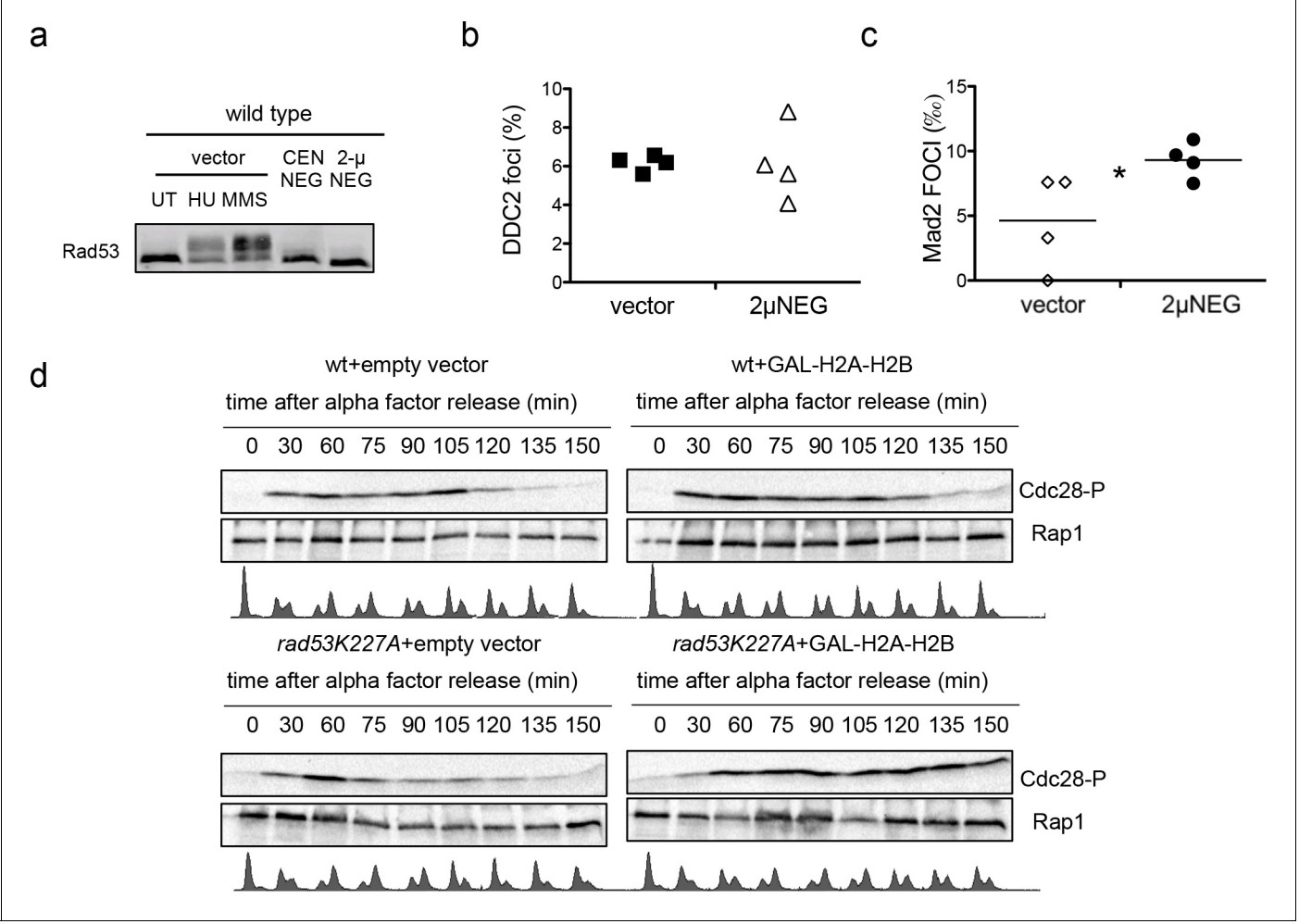

**Figure 7.** Plasmid-driven overexpresson of histones promotes Cdc28$^{CDK1}$ phosphorylation. (**a**) Rad53 phosphorylation in wild type cells transformed with an empty vector (UT), the centromeric, or the 2μ version of *HTA1-HTB1ΔNEG* (*CENΔNEG* and *2μΔNEG* respectively). Phosphorylated Rad53 appears as a shifted band clearly observed after treatment with 200 μM Hydroxyurea (HU) or 0.03% MMS (MMS). (**b**) Average and SEM (four independent experiments, 200 cells counted for each) of wild-type cells exhibiting Ddc2-Foci after transformation with an empty vector or the *2μΔNEG* vector. DNA content was measured for each sample to confirm that cells still remain haploid (not shown). (**c**) Average number of Mad2 foci (Mad2-GFP) that co-localise to kinetochores (Mtw1-mCHERRY) in wild-type cells after transformation with an empty vector or the *2μΔNEG* vector. DNA content was measured for each sample to confirm that cells still remain haploid (not shown). p-value=0.031 (t-test, paired, two tails) (**d**) Phosphorylation of Cdc28 upon H2A-H2B overexpression. Wild type and *rad53K227A* cells were transformed with either a control vector or a high-copy vector that expresses histones H2A and H2B from a galactose-inducible promoter. Cells were grown in raffinose, arrested in G1 with alpha-factor, incubated 2.5 hr with galactose to induce H2A-H2B expression, and released from alpha-factor block to follow cell cycle progression (FACS) and Cdc28 phosphorylation. Alpha factor was re-added 75 min after to re-arrest cells in G1. These experiment was performed twice and gave similar results.

DOI: https://doi.org/10.7554/eLife.35337.024

The following source data is available for figure 7:

**Source data 1.** Plasmid-driven overexpression of histones promotes Cdc28CDK1 phosphorylation
DOI: https://doi.org/10.7554/eLife.35337.025

with an empty vector (*Figure 8—figure supplement 1d*). These deletions had almost no impact on the the cell cycle arrest observed in *rad53-AID-lsm1Δ*-treated cells individually (*Figure 8—figure supplement 1d*) but significantly suppressed the arrest when combined (*Figure 8b* and *Figure 8—figure supplement 1d*). Overall, our results confirm that the WGD events observed upon histone overexpression are not generated by a persistent activation of the DDR, the SAC, or a stabilisation of Swe1$^{WEE1}$. The results observed with Mad2 and Swe1$^{WEE1}$ suggest that the SAC might instead cooperate with Swe1$^{WEE1}$ and help to prevent them.

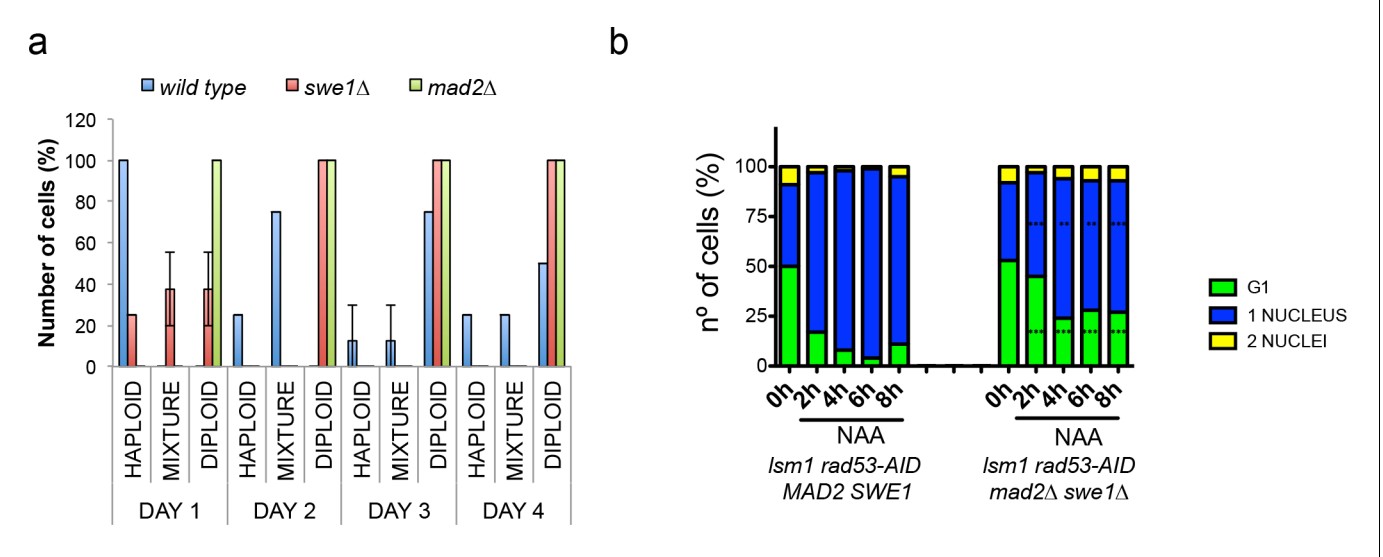

**Figure 8.** Deletion of *SWE1* or *MAD2* enhances WGDs. (a) Percentage of cells with a haploid, mixed, or diploid DNA content during a 4 days (D1–D4) time-course in wild type, *swe1Δ*, or *mad2Δ* cells after transformation with the *2μΔNEG* vector. These kinetics were performed as the ones shown in *Figure 2a*. Eight clones were analysed for each genetic background. DNA content of the indicated mutant cells is measured after transformation with 2μΔNEG. Samples were taken every 24 hr and analysed by FACS to estimate DNA content. Eight independent clones were analysed for each genetic background. (b) Number of unbudded and budded cells with one or two well-differentiated nuclei observed in *rad53-AID lsm1Δ* and *rad53-AID lsm1Δ mad2Δ swe1Δ* cells after NAA treatment. Percentages were estimated by DAPI in fixed samples. At least 100 cells were counted. p-Values were obtained for each cell type with a one-way ANOVA test (**p<0.01 ***p<0.001) in which two independent experiments were compared.

DOI: https://doi.org/10.7554/eLife.35337.026

The following source data and figure supplement are available for figure 8:

**Source data 1.** Deletion of SWE1 or MAD2 enhances WGDs.

DOI: https://doi.org/10.7554/eLife.35337.028

**Figure supplement 1.** Deletion of SWE1 or MAD2 enhances WGDs.

DOI: https://doi.org/10.7554/eLife.35337.027

## Discussion

In this study, we demonstrate that high levels of histones interfere with chromosome segregation and are able to promote WGDs. Collectively, our results indicate that these events are linked to a defect in the incorporation of the histone variant Htz1[H2A.Z]. We propose a competition model in which the canonical histone H2A competes with the histone variant H2A.Z. In this model, increasing the levels of canonical histone H2A will decrease the ratio H2A.Z/H2A in nucleosomes thereby decreasing the interaction of H2A.Z with the readers of this histone variant. Histone overexpression has been previously shown to saturate certain histone-modifying enzymes (*Singh et al., 2010*) and could also in theory, saturate chromatin remodelers able to bind and exchange H2A by H2A.Z. However, this explanation is not sufficient to explain how the overexpression of histones H2A and H2B trigger WGDs since H3 and H4 overexpression also triggers WGDs in a *rad53K227A* mutant. Two reports have shown that the incorporation of Htz1[H2A.Z] (*Ranjan et al., 2013*) and condensin (*Toselli-Mollereau et al., 2016*) requires nucleosome-free regions to be incorporated efficiently to chromatin (*Ranjan et al., 2013*). Our results show that the regions in which Htz1[H2A.Z] incorporation is decreased in *rad53-AID-lsm1Δ*-treated cells exhibit higher nucleosome occupancy and are less accessible to Micrococcal Nuclease digestion. Accumulation of histones beyond replication might have a general effect on chromatin structure and reduce the amount of nucleosome-free regions. This general effect on chromatin structure may explain why the overexpression of Htz1[H2A.Z] is able to decrease the amounts of WGDs in cells overexpressing histones H2A and H2B but does not improve the incorporation of Htz1[H2A.Z] in *rad53-AID-lsm1Δ*-treated cells. We infer from our results that histone expression may be restricted to S-phase to avoid an excessive incorporation of histones to chromatin and maintain nucleosome-free regions that would in particular promote the specific

incorporation of Htz1$^{H2A.Z}$ (*Boyarchuk et al., 2014*; *Nekrasov et al., 2012*). Efficient incorporation of Htz1$^{H2A.Z}$ would regulate condensin recruitment throughout the cell cycle (*Figure 9*). Condensin depletion has been previously linked to chromosome segregation defects and WGDs (*Oliveira et al., 2005*; *Woodward et al., 2016*) and could explain why high levels of histones promote WGDs.

We have also addressed in this study if high levels of histones modulate cell cycle progression to avoid their potential toxicity. Our results confirm that histone overexpression does not trigger the canonical DNA damage response. The Pds1 stabilisation observed in *rad53-AID lsm1Δ* treated cells added to the fact that *MAD2* deletion enhances WGDs indicate that the SAC could be activated in cells that overexpress histones. This activation, however, does not seem to be the cause behind WGDs but rather a mechanism that helps to prevent them. One of the most interesting results of our checkpoint analysis is that cells that accumulate large amount of histones exhibit a DNA damage-independent phosphorylation of Cdc28$^{CDK1}$ that likely results from the stabilisation of Swe1$^{WEE1}$. As mentioned, this activation would not be the cause of WGDs but rather would help to prevent them. Swe1$^{WEE1}$ was previously reported to interact with histone H2B in yeast and humans, phosphorylate H2B at Tyr$_{37}$ (Tyr$_{40}$ in yeast), and promote the efficient repression of histones transcription at the end of S-phase (*Mahajan et al., 2012*). The fact that this kinase is stabilised in the presence of high levels of histones unveils an unsuspected and general role of Swe1$^{WEE1}$ in histone metabolism. Noteworthy, low levels of histone supply during S-phase have been shown to delay mitosis in *Drosophila* embryos through the increase of CDK1 phosphorylation levels mediated by the transcriptional downregulation of the Cdc25 phosphatase (*Günesdogan et al., 2014*). Our results added to this study suggest that the regulation of Cdc28$^{CDK1}$ could be a mechanism by which cells can respond to both high and low levels of histones.

WGDs are quite frequently observed in cancer (*Zack et al., 2013*) and play an important role in tumorigenesis (*Davoli and de Lange, 2012*; *Dewhurst et al., 2014*; *Gordon et al., 2012*; *Janssen and Medema, 2013*; *Santaguida and Amon, 2015*). Given the different ways in which higher and lower eukaryotes regulate histone levels, one reasonable question that arises from our study is if our results can be extrapolated to higher eukaryotes. Overexpression of histone H2A has been linked to the transformation of normal liver to preneoplastic and neoplastic stages of Hepatocellular Carcinoma (*Khare et al., 2011*) where WGDs and aneuploidy are both frequent (*Davoli and de Lange, 2011*; *Duncan et al., 2010*). Two different lines of investigation, one in drosophila and one in bronchial cells, have shown that a partial depletion of the histone mRNA-binding factor SLBP leads to the accumulation of abnormal levels of polyadenylated histones mRNA beyond replication. These polyadenilated mRNAs can be translated and escape the normal cell cycle control of histones (*Brocato et al., 2015*). In the first study, this depletion correlated to the generation of a tetraploid population (*Salzler et al., 2009*). The second study demonstrated that arsenic, a common carcinogenic agent able to promote aneuploidy and WGDs, partially depletes SLBP, which in turn increases the tumorigenic potential of transformed cells (*Brocato et al., 2015*; *States, 2015*). Interestingly, arsenic-induced cellular transformation induces a more compact chromatin structure (*Riedmann et al., 2015*). Our work, combined with these reports, supports the idea that histone deregulation might constitute an important and yet unexplored potential source of genome instability in cells. It would therefore be interesting to revise upwards the amplification of the small arm of chromosome 6, which contains 55 out of the 65 genes that encode canonical histones. This amplification takes place in a wide variety of tumors and correlates with cancer aggressiveness (*Santos et al., 2007*).

## Materials and methods

### Strains, plasmids, growth conditions and cell-cycle synchronisation

All strains used in this study derive from the S288C background. Genotypes are listed in *Supplementary file 1a*, plasmids in *Supplementary file 1b* and oligonucleotidess in *Supplementary file 1c*.

Rad53 Auxin-Induced Degron (AID) was constructed as described in *Nishimura et al., 2009*. Rad53 was tagged by PCR using a carboxy terminal tag obtained from vector pMK43. The strain carrying the *rad53-AID* allele also bears the integrative vector pNHK36, which contains the OsTYR1

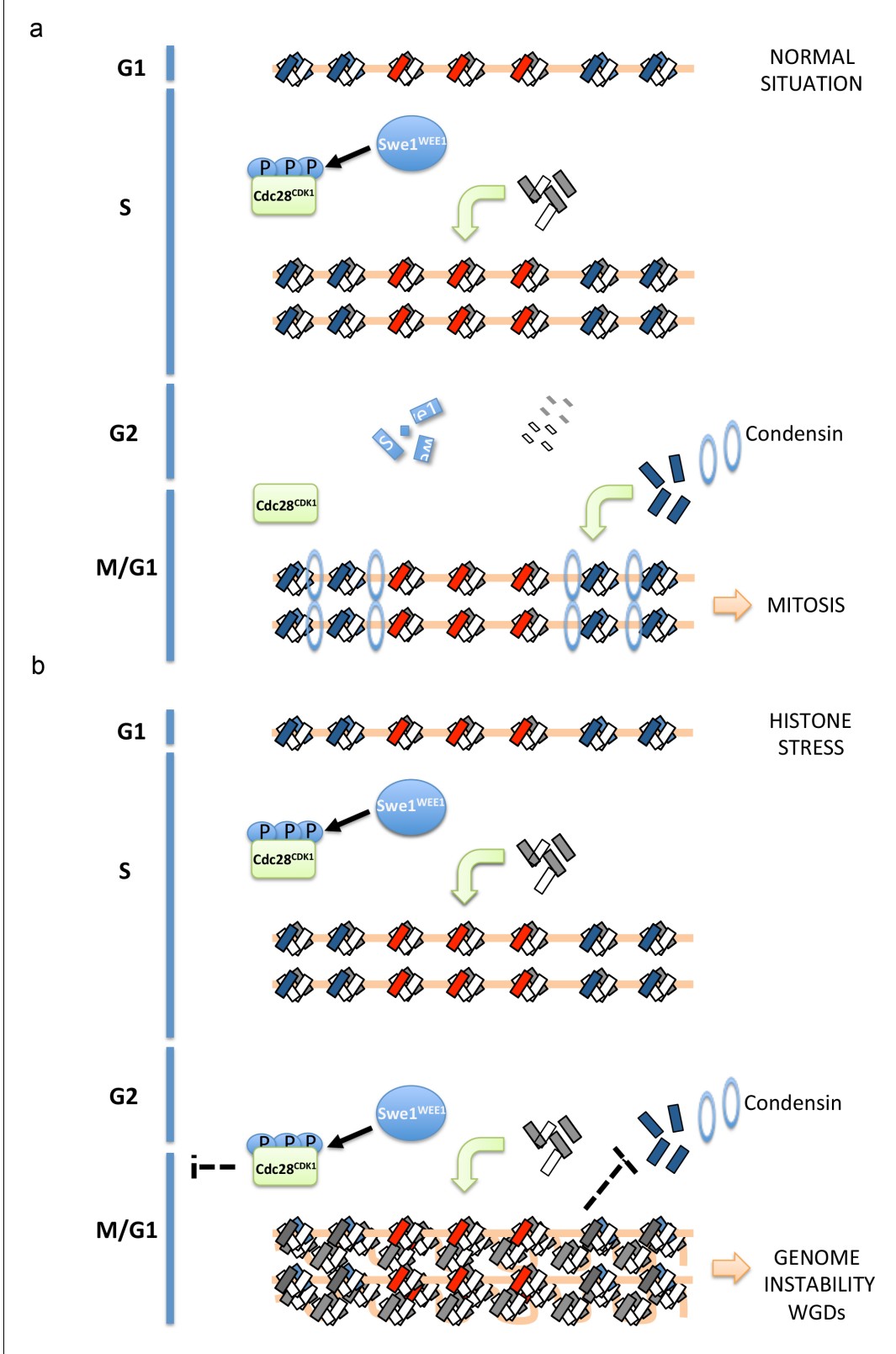

**Figure 9.** Proposed model to explain how histone-stress impacts mitosis. Histone dimers are depicted as small rectangles (H3-H4 in white, H2A-H2B in grey and H2A.Z-H2B in blue). During an unperturbed cell cycle (a, upper scheme), canonical histones and Swe1$^{WEE1}$ increase during replication. Swe1$^{WEE1}$ will phosphorylate Cdc28$^{CDK1}$ and maintain it inactive. During G2, histone synthesis will be repressed and all histones (mRNAs and proteins) that are not incorporated to chromatin as well as Swe1$^{WEE1}$ will be degraded. During mitosis, Htz1$^{H2A.Z}$ will stabilise condensin recruitment at

*Figure 9 continued on next page*

*Figure 9 continued*

pericentromeric regions allowing the proper function of this complex in chromosome segregation. When histone degradation is compromised (b, lower scheme), cells reach G2 with high levels of histones (histone-stress). This accumulation of histones will promote Cdc28$^{CDK1}$ phosphorylation and inactivation. This inhibition will delay the entry into mitosis and presumably give time to the cell to lower histone levels. Since this phosphorylation depends on Swe1$^{WEE1}$, we propose that histone-stress promotes Cdc28$^{CDK1}$ phosphorylation through a stabilisation of Swe1$^{WEE1}$. Cells unable to efficiently lower histone levels after replication will increase the amount of canonical nucleosomes incorporated at centromeres and pericentromeres. We propose that this increase in nucleosome density will decrease the efficient exchange of histone H2A by histone Htz1$^{H2A.Z}$, and reduce Htz1$^{H2A.Z}$ incorporation. This defect in incorporation would consequently lead to a less stable association of condensin to pericentromeres and trigger chromosome segregation defects.

DOI: https://doi.org/10.7554/eLife.35337.029

protein under the control of the *GAL1*-inducible promoter. Both vectors were obtained from Addgene. All *rad53-AID* strains were constructed in an *sml1Δ* background to compensate the essential function of Rad53 in promoting dNTP production. The functionality of tagged proteins was systematically checked.

Exponentially growing *rad53K227A* (or *tom1Δ)* cells were transformed with a centromeric vector carrying the *HTA1-HTB1ΔNEG* construct while wild-type cells were transformed with a 2-μ vector carrying the same cassette. After transformation, strains were plated on SC-Uracil selective media at 30°C. The frequency of WGDs in these strains was calculated based on the percentage of cells able to form a colony after 3 days of growth compared to the total amount of colonies visible after 5 days of growth. This method to estimate WGDs is based on the growth advantage that WGDs confer upon histone overexpression. Ploidy analysis of individual transformants by FACS indicates that roughly than 90% of cells able to grow at day 3 have experienced a WGD (8 or 9 out of 10, two independent repeats for each strain). As control, similar experiments were performed with strains transformed with an empty vector to show that generation of WGDs in linked to the presence of the *ΔNEG* construct. Time courses to determine the timing of WGDs were performed always from single small colonies that were grown and maintained in exponential growth conditions in liquid SC-uracil. As control, single colonies obtained from the corresponding strain transformed with an empty vector were analysed in parallel. Samples were taken every 24 hr to estimate the proportion of haploids and diploids by FACS.

For plate growth assays, yeast cultures were diluted to an O.D.$_{600nm}$ of 0.5 and serial 1:10 dilutions were spotted on the indicated plates. Doubling times were calculated from 12 hr time-courses in which cells were maintained in exponential growth conditions. Doubling times correspond to the growth rate of cells that still remain haploid during the time course (checked by FACS).

For experiments involving the rad53-AID construct, glucose was replaced with galactose to allow the expression of the OsTyr1 protein, required for AID-mediated degradation. These experiments were always carried out in yeast-peptone rich media (YP) except those performed for microscopy that were carried in SC media. Rad53 depletion was induced adding 1-Naphthaleneacetic acid (SIGMA N0640, referred as NAA or Auxin in the text). NAA was normally added to a final concentration of 500 μM (rich media) or 200 μM (SC) unless indicated. This difference in NAA concentration used between rich and minimal media is due to the fact that high concentrations of NAA in minimal media have deleterious effects on yeast growth.

Cell-cycle synchronisations in G1 were performed adding alpha factor to a final concentration of 0.01 μg/ml. All strains were *bar1Δ* to ensure a complete arrest that was checked by FACS and microscopy (90% or more of G1-arrested cells). Cell-cycle synchronizsations in G2/M were performed adding 10 μg/ml of Nocodazole. Cells were maintained in the presence of alpha-factor or Nocodazole for a duration corresponding of at least one whole cell cycle (estimated from duplication time). Cell-cycle arrest was confirmed by FACS and DAPI staining after fixation in 70% ethanol. After rehydratation in 1X PBS, samples for FACS were incubated for 3 hr with RNaseA (0.5 mg/ml) at 37°C. Cells were then resuspended in 0.5 μg/ml SYTOX green in PBS and incubated for at least 20 min at room temperature. After a 1X PBS wash, cells were briefly sonicated to remove cell clumps and DNA content was determined with a Becton Dickinson FACS-Calibur. For DAPI analysis, fixed cells were incubated with 0.2 μg/ml of DAPI for 30 min before washing twice with 1x PBS. Unless indicated, three independent repetitions were performed for all experiments.

## Western blot, chromatin immunoprecipitation, MNase digestion, histone purification and pulse field gel electrophoresis

Cell lysates to perform Western Blot were prepared using a standard protocol of TCA precipitation. Laemmli-boiled crude extracts were run on a SDS-polyacrylamide gel (8, 10 or 12% depending on experiment) and transferred to a nitrocellulose membrane (Hybond-ECL). Membranes were blocked in PBS-T milk 5% and incubated with antibodies: 9E10 Myc (sc-40, Santa Cruz Biotechnology), anti-Rad53 (mouse monoclonal gift from Marco Foiani), HA (clone 3F10, Roche), H3 (ab1791, Abcam), H2B (ab1790, Abcam), H2A (39235), H4 (ab10158, Abcam), H2A.Z (39647 Actif Motif), phospho-cdc2 (Tyr15) (9111 Cell Signalling), cdc2-p34 (sc53, Santa Cruz Biotechnology), Clb2 (SC9071, Santa Cruz Biotechnology), Actin (ab8224, Abcam) Swe1 (rabbit polyclonal, gift from Doug Kellogg) and Rap1 (V. Géli Lab). Peroxidase-conjugated goat anti-mouse or anti-rabbit IgG (both from Bio-Rad) were used to detect proteins using a ChemiDoc Gel Imaging System. Protein quantification was performed with the ImageJ software. For Chromatin Immunoprecipitation (ChIP) and MNase digestion samples were previously fixed 15 min with 1% formaldehyde. Glycine was added to quench the reaction at a final concentration of 125 mM. Cells were sedimented, washed twice with cold TBS and stored at $-80°C$ until use. ChIP samples were obtained breaking cells with a FastPrep-24 (116004500 MP biomedicals) (3 pulses of 30 s at 5.5 intensity) in lysis buffer (HEPES 50 mM NaCl 140 mM 1 mM EDTA 1%Triton 0.1% Sodium deoxycolate 1 mM PMSF) supplemented with protease cocktail inhibitors (ROCHE). Supernatant was transferred to new tubes by centrifugation piercing the bottom of the tube with a G25 needle and chromatin was concentrated by centrifugation. Chromatin was sheared via sonication to a size between 200 and 600 bp (Diagenode) in a bioruptor (Diagenode). The supernatant was divided in equal volumes, keeping 10 μl that were used as the input DNA control, incubated with specific antibodies (4°C overnight incubation, 1 μg of antibody) and immunoprecipitated with Protein G Dynabeads (Novex). After immunoprecipitation, samples were washed once with 1 ml of the following solutions: lysis buffer, lysis buffer 0.5M NaCl, wash buffer (0.25M LiCl 10 mM Tris HCl 1 mM EDTA 0.5% NP-40 0.5% Sodium deoxycolate) and TE. Samples were then eluted from magnetic beads with a 1% SDS TE solution, incubated overnight at 65°C to reverse crosslinking, treated with 0.15 mg of Proteinase K, and extracted using a DNA purification kit (Qiagen 28004). DNA was analysed by real-time qPCR using SYBR Green Premix Ex Taq (Takara) in a Rotor Gene 6000 (Corbett Research, Labgene, Archamps, France). Primers used are listed in *Supplementary file 1c*. Extracts for Mnase were resuspended in 1M sorbitol, digested 1 hr with 4.5 mg of Zymoliase 20T (AmsBio 120491–1) and treated 30 min with different concentrations of Mnase (SIGMA N3755) that was inactivated with SDS (0.4%) and EDTA (8.5 mM). Samples were incubated 1 hr and 30 min at 37°C with proteinase K and overnight at 65°C for reversal crosslinking. DNA was extracted from samples using a standard phenol-chloroform extraction, treated with RNAse and loaded in a 2% agarose gel. Histones were purified as described (*Jourquin and Géli, 2017*). Samples for PFGE were obtained as previously described (*Dueñas-Sánchez et al., 2010*). PFGE was performed on a BioRad CHEF DR-III system in a 0.8% agarose gel in 1 × TAE (40 mM Tris-acetate buffer, 2 mM Na$_2$EDTA, pH8.3) at 14°C. After electrophoresis, DNA was stained and visualised with Ethidium Bromide.

## RNA extraction and Q-PCR analysis

RNA was extracted using a standard protocol with hot acid phenol. RNA samples were treated with DNase I (USB) prior to use in RT-PCR experiments. Quantitative RT-PCR experiments were performed using one-step RT-PCR, in a LightCycler 480 II (Roche) using One Step SYBR PrimeScript RT-PCR Kit (Takara Bio Inc., Japan), 0.2 μM of each primer and 50 ng of RNA in a 10 μl reaction. Primers used: H2B (HTB1) HTB1-F: AGAGAAGCAAGGCTAGAAAGGA HTB1-R: GGAAATACCAGTG TCAGGGTG; TUB1 TUB1-F: TCTTGGTGGTGGTACTGGTT TUB1-R: TGGATTTCTTACCGTA TTCAGCG.

## Microscopy

All microscopy analyses except those corresponding to live-microscopy were performed in liquid SC synthetic media using a Nikon Eclipse Ti microscope with a 100x objective. Cell images were captured with a Neo sCMOS Camera (Andor). Images were analysed using ImageJ on 2D-maximum projections from 11-Z-stacks spaced 0.5μ each. For live microscopy, cells were plated in SC synthetic

medium on concanavalin A–coated (C2012, Sigma) Lab-Tek chambers (Thermo Fisher Scientific). Imaging was performed using a spinning-disk confocal microscope (Revolution XD; Andor Technology) with a Plan Apochromat 100×, 1.45 NA objective equipped with a dual-mode electron-modifying charge-coupled device camera (iXon 897 E; Andor Technology). Time-lapse series with variable stacks (check figure legends) were acquired every 5 min. iQ Live Cell Imaging software (Andor Technology) was used for image acquisition. Images were analysed on 2D maximum projections and denoised with the Despeckle function in ImageJ 1.46b (National Institutes of Health). Graphs and statistical analysis were performed with Prism or Excel. Fun1 staining analysis was performed with a commercial kit (L-7009 from Molecular Probes) following the standard protocol described in the kit.

## Acknowledgements

We thank A Gunjan, J Tyler, S Biggins, F Prado, Doug Kellogg, and S Mahajan for sharing strains, plasmids and/or reagents; We would also like to thank C Machu for imaging expertise and JH Guervilly, Felix Prado, E Bailly and V Geli team members for discussions. DM was supported by a post-doctoral fellowship from the Association pour la Recherche sur le Cancer (Fondation ARC). Work in VG laboratory is supported by the"Ligue contre le Cancer' (Equipe Labéllisée 2017). Work in the laboratory of M M is supported of the European Research Council (ERC) Starting Grant 2010-St-20091118, and the Spanish Ministry of Economy and Competitiveness BFU2012-37162 to MM, and 'Centro de Excelencia Severo Ochoa 2013–2017', SEV-2012–0208 to the CRG. S C is supported by grants BFU2013-48643-C3-1-P from the Spanish MiNECO, and P12-BIO1938MO from the Regional Andalusian Government, both incluidng European Union funds (FEDER).

## Additional information

### Funding

| Funder | Grant reference number | Author |
| --- | --- | --- |
| Ligue Contre le Cancer | Equipe Labéllisée 2017 | Vincent Geli |
| Fondation ARC pour la Recherche sur le Cancer | Aide Posdtdoctorale | Douglas Maya Miles |
| Ministerio de Economía y Competitividad | BFU2012-37162 | Manuel Mendoza |
| European Research Council | 2010-St-20091118 | Manuel Mendoza |
| Regional Andalusian Government | P12-BIO1938MO | Sebastian Chavez |
| Ministerio de Economía y Competitividad | BFU2013-48643-C3-1-P | Sebastian Chavez |

The funders had no role in study design, data collection and interpretation, or the decision to submit the work for publication.

### Author contributions

Douglas Maya Miles, Conceptualization, Investigation, Supervision, Validation, Writing—original draft; Xenia Peñate, Investigation, Validation; Trinidad Sanmartín Olmo, Investigation, Methodology; Frederic Jourquin, Maria Cruz Muñoz Centeno, Investigation; Manuel Mendoza, Investigation, Methodology, Writing—review and editing; Marie-Noelle Simon, Investigation, Writing—review and editing; Sebastian Chavez, Conceptualization, Writing—review and editing; Vincent Geli, Conceptualization, Investigation, Writing—review and editing

### Author ORCIDs

Douglas Maya Miles  https://orcid.org/0000-0002-0669-6526
Xenia Peñate  https://orcid.org/0000-0002-4117-888X
Vincent Geli  https://orcid.org/0000-0002-4103-7462

**Decision letter and Author response**
Decision letter https://doi.org/10.7554/eLife.35337.034
Author response https://doi.org/10.7554/eLife.35337.035

## Additional files

### Supplementary files
• Supplementary file 1. Supplementary files 1a-c.
DOI: https://doi.org/10.7554/eLife.35337.030

• Transparent reporting form
DOI: https://doi.org/10.7554/eLife.35337.031

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
