## [Decision Letter]

[Editors’ note: this article was originally rejected after discussions between the reviewers, but the authors were invited to resubmit after an appeal against the decision.]

Thank you for submitting your work entitled "A histone-sensing checkpoint prevents undesired endomitosis promoted by histone overexpression" for consideration by *eLife*. Your article has been reviewed by three peer reviewers, and the evaluation has been overseen by a Reviewing Editor and a Senior Editor. The reviewers have opted to remain anonymous.

Our decision has been reached after consultation between the reviewers. Based on these discussions and the individual reviews below, we regret to inform you that your work will not be considered further for publication in *eLife*.

The reviewers were excited by the initial observations in the manuscript. However, they felt that there were too many shortcomings in the data and unexplored mechanistic issues that the current manuscript is not sufficient for publication. They also felt that these issues could not be addressed in a timely manner and did not recommend revision of the manuscript for *eLife*. We hope that the detailed reviews below will be useful for moving the study forward.

*Reviewer #1:*

Miles et al. present multiple lines of evidence that histone overexpression causes mitotic defects in yeast cells. The DNA damage and spindle assembly checkpoints are not activated by such a stress, whereas Tyr_15_ of Cdk1 is phosphorylated, suggesting a novel Cdk1-dependent, but DDR- and SAC-independent regulatory function of mitosis. Whole genome duplication is an important phenomenon associated with development of many species and of human cancers. This work is thus of strong interest to many. However, this manuscript suffers from a slew of deficiencies, ranging from writing to experimental design to result interpretation. It'll take a very significant effort to resolve these issues satisfactorily.

1) One of the most interesting and valuable observations from this work is probably the visualization of how diploids might be generated in the wake of histone overdose. That is, the movement of both sets of duplicated genome to the daughter cell without cytokinesis. However, the authors fall short on testing the functionality of other key mitotic machinery such as spindle, spindle pole bodies, separation of centromeres, etc. of these cells. Moreover, there is no telling whether these cells can proceed successfully to the next round of mitosis, resulting in the formation of a homogenous diploid population.

2) It is good that the authors use different approaches to induce histone stress, however, the basis of which remains unclear. The authors cite a previous finding that Swe1 contacts H2B, but do not elaborate. They also suggest that reduced H2A.Z recruitment to chromatin might be responsible. However, there is no report of a similar mitotic defect in known htz1 alleles. It is also unclear how skewing histone expression would lead to WGD. For example, what is the minimal degree of overexpression needed? Figure 1 shows that GAL-controlled expression of all four core histones in the same *rad53K227A* background results in the retention of 1n population, suggesting that concerted expression of histones, not merely the number of histone molecules, is critical for mitotic integrity. Furthermore, the authors favor the idea of downregulation of H2A.Z loading to pericentromeres being the underpinning of the observed mitotic defects. If so, will co-expressing H2A.Z with H2A and H2B alleviate at least some of the stress? How is it that H3/H4 overexpression also confers the growth phenotype as does H2A/H2B ectopic expression?

3) The ChIP-qPCR data are weak. What is the resolution of the qPCR? What are the reaction conditions? The provision of oligo primers in Table 3 does not help much.

4) Figure 7 MNase assays suggest that the subject chromatin is more resistant to MNase, indicating higher nucleosome occupancy. However, this single-dose experiment does not afford sufficient quantitative power. There are many advanced, more sophisticated approaches available for MNase assays. The authors should go deeper than the current endeavor.

5) The experimental procedures provided are far from complete. Key experiments such as MNase digestion, cell lysate preparation for immunochemical analysis, and ChIP-qPCR are all left out.

6) There are many writing and grammatical errors throughout the text.

*Reviewer #2:*

Genome stability is critical for normal physiology of an organism. Genomic instability at the chromosome level results in aneuploidy, which can cause birth defects and cancer. As cancer cells typically gain copies of chromosomes, one popular theory is that aneuploidy follows whole-genome duplication and polyploidy. Polyploidy puts more burden on the mitotic chromosome segregation machinery, increases errors in that process, and allows cells to better tolerate the resulting aneuploidy. Thus, it is important for us to understand the potential causes and mechanisms of polyploidy. Using the budding yeast as a model organism, Miles et al. showed in the current study that histone overexpression (coupled with or without mutations in other histone-surveillance genes) causes polyploidy. They further probed the underlying mechanisms and made two interesting findings. First, histone overexpression promotes Wee1-dependent negative phosphorylation on Cdk1 and delays mitotic entry. This mechanism is not, however, required for polyploidization, but may impede it. Second, histone overexpression limits the deposition of the histone variant H2A.Z and its downstream effector condensin on chromosomes, particularly near centromeres. They believe that this mechanism underlies polyploidization caused by histone overexpression.

Overall, this study advances our understanding of ploidy regulation. The findings are novel and significant. The results are for the most part convincing and logically presented. Publication is recommended, provided that the authors address the following major points.

1) While it is clear that H2A.Z and condensin deposition is defective under conditions of histone overexpression, the evidence linking this effect to polyploidization needs to be further strengthened. The only strong evidence presented is that deletion of the H2A.Z depositor Swr1 exacerbates the effect of histone overexpression. A genetic suppression experiment will be much more convincing. The authors should test whether H2A.Z overexpression or forced targeting of condensin to peri-centromeres reduces polyploidy caused by histone overexpression.

2) Much of the results on DNA damage checkpoint, spindle assembly checkpoint, and Shugoshin are negative results. These sections should be shortened. This will leave more space for the authors to address point 1 above.

*Reviewer #3:*

This study builds on the notion that histone protein levels are tightly regulated by multiple mechanisms and that misregulation or histone overexpression causes toxicity, cell cycle arrest, sensitivity to DNA damaging agents, and mitotic chromosome loss.

One key observation in this study is that conditions that cause increased histone protein expression lead to whole genome duplication events in yeast, i.e. cells fail to divide the nuclear material between mother and daughter cell in a timely manner; these cells stop proliferating, or the resulting diploid cells as a result of endoreduplication have a growth advantage and rapidly take over the population. This is also apparent on plates where diploids form larger colonies. Upon presumed severe increased histone levels (by disrupting the histone RNA and protein degradation pathways, which is lethal) cells arrest in G2/M and fail to exit from a G2/M arrest. In this first part, the authors investigated whether there is a sensing mechanism, or a checkpoint, for high histone levels. The failure to exit from G2/M was accompanied by a delay in cohesin degradation and increased Cdc28 phosphorylation by Swe1. The authors propose that this event delays the entry into mitosis. However, deletion of Swe1 abolishes Cdc28 phosphorylation but does not suppress the G2/M arrest. Swe1 loss does enhance WGD events.

In the second part, investigating the cause of the WGD events, the authors show that presumed very high histone levels do not affect CENP-A recruitment to centromeres, but do impair H2A.Z and condensin recruitment to pericentric heterochromatin. A condensin mutation enhances the toxicity of histone overexpression and loss of the H2A.Z assembly factor Swr1 enhances the selection for diploid cells upon transformation with a histone H2A/B overexpression vector, together suggesting that loss of these two chromatin components can further exacerbate the effects of increased histone dosage. This leads the authors to suggest that the decrease in H2A.Z and condensin is at least one of the causes that trigger WGDs when excess free histones accumulate. However, another interpretation of these results (in essence negative genetic interactions) is that histone excess and loss of H2A.Z/condensin act by independent mechanisms because a loss of H2A.Z alone was not sufficient to cause WGD events. It is likely that increased histone dosage affects WGD also by other mechanisms.

This study presents a series of interesting observations that warrant further investigation. However, the claims made by the authors are not always justified and supported by data. Importantly, whether there is a CDC28/Swe1-mediated sensing mechanism has not been fully addressed. In addition, for many of the experiments, replicates are missing and the study would benefit from a more quantitative appreciation of the altered histone levels. Finally, the story is composed of two pieces that are currently not well connected.

- The claims made by the authors are not always justified and supported by data, e.g. "we have uncovered a new type of stress (histone-stress) able to trigger a stabilization of Swe1^WEE1^ that promotes Cdc28^CDK1^ phosphorylation and delays the entry into mitosis". Similar conclusions are drawn elsewhere (e.g. Discussion, third paragraph and Abstract). That histone overexpression causes stress was already known. That it stabilizes Swe1 is not shown in this manuscript. Furthermore, that Cdc28 phosphorylation delays entry into mitosis is not so clear because deletion of Swe1, the responsible kinase, does not affect the delay.

- For many key experiments replicates or quantification of replicate experiments are missing. For most of the western blots, FACS analyses, microscopy, and MNase sensitivity assays in the manuscript no replicates or quantification of replicates are provided. The authors should provide additional data on these experiments to demonstrate reproducibility, following the guidelines of *eLife*.

- The authors use various conditions to increase histone expression outside S-phase. One approach is overexpressing the histone genes from strong promoters. This is in principle straightforward. The other main approach is combining a *Rad53* degron with an *lsm1* mutation, which causes cell death. There are several issues with the approaches.

1) In the latter case (*lsm1*+*rad53*-aid) it cannot be excluded that the phenotypes seen are at least in part caused by other functions of the inactivated proteins. Although this is partially addressed in Figure 1, the Cdc28 phenotype has only been observed in *rad53* mutant backgrounds.

2) The authors use various conditions to induce histone overexpression but show in very few cases to what extent histone proteins are overexpressed. This is an important issue because the various methods and combinations of alleles and/or plasmids have different phenotypic consequences (e.g. 2-micron-δ NEG vs. *lsm1*+*rad53*-aid). The authors should examine histone protein levels for the different conditions used, for example not only *lsm1* with or without *rad53*-aid (Figure 3), but also for the conditions used in Figure 1 and Figure 5. For example, based on the phenotypes, one might assume that the δ NEG or GAL-H2A-H2B plasmids cause lower histone protein expression than the combined *rad53*+*lsm1* mutation. However, this is not demonstrated by histone blots.

3) The assay first shown in Figure 2 involves transformation with a 2-micron-HTA-HTB1-δ-NEG plasmid. The authors monitor diploidization of the pool of cells over time starting at the time of transformation. This is a somewhat unconventional assay that would benefit from more explanation of the approach and experimental details. For example, at what time point was the vector control analyzed? It would be appropriate to take a vector control along for every time point and also take a t=0 point. In this assay several factors can influence the outcome or dynamics: First, after transformation only a small proportion of the cells will contain a plasmid and survive the selection, most cells will die. Therefore, the kinetics might depend on the transformation efficiency. Second, cells that become diploid will have a growth advantage and take over the population. How fast they do so will depend on the rate of diploidization as well as the growth rate. These different parameters should be explained more clearly and taken into account experimentally. Another point here concerns panel 2D: what is the time point after transformation, how did the authors assess whether the cells examined actually contained a histone plasmid, and what are the numbers for a wild-type cell or vector control?

Other Points:

Figure 1. How long were the cells grown in galactose to induce histone expression?

The authors state that '*Rad53* depletion was lethal in the absence of *Lsm1* and largely increased the amount of histones in the cell (Figure 3).' However, the increase in H2B seems modest.

In Figure 3, the colony formation data should be represented as absolute numbers instead of relative

In Figure 5, the blots are confusing. In the top panel it is hard to judge the absence of the band in the last lane, in the bottom panel it is not clear why the intensity of the two bands changes?

In Figure 5, the cell cycle progression is hard to judge from the images shown; the phenotype of the *rad53* mutant seems independent of the histone plasmid (and this may also be the case for cdc28 phosphorylation). Furthermore, in the third paragraph of the subsection “Histone accumulation triggers a Cdc28^CDK1^ phosphorylation that delays the entry into mitosis” the authors conclude that the phosphorylation in the upper right panel was slightly increased. However, this cannot be concluded from the blots shown because the comparison is made between two separate blots.

For Figure 7–Figure 8 it is not clear how many biological replicates were performed.

The authors suggest that '*rad53 AID lsm1*Δ treated cells were more resistant to MNase digestion' (Figure 7, and subsection “High levels of histones decrease Htz1^H2A.Z^ and condensin incorporation to pericentromeric chromatin”, second paragraph). This is not immediately obvious from the figure. The authors should provide additional analyses or discussion to support this conclusion.

Regarding the mechanism of WGD by increased histone dosage, a role for H2A.Z in condensin stability has previously been shown in *S. pombe*. Given the differences in pericentric chromatin between the models yeasts, it would be useful to establish whether there is also a connection in *S. cerevisiae*.

[Editors' notes: What follows is the decision after the authors appealed the previous decision.]

Thank you for choosing to send your work entitled "A histone-sensing checkpoint prevents undesired endomitosis promoted by histone overexpression" for consideration at *eLife*. Your article and your letter of appeal have been considered by a Senior Editor, and we regret to inform you that we are upholding our original decision.

All three reviewers think the work is of sufficient interest, in principle, for publication in *eLife*. However, the reviewers pointed out many problems, and Reviewers 1 and 3 in particular thought that addressing these problems would take too much time to merit a "Revise" decision. I should say that *eLife* policy is that "revise" decisions should be made when the amount of time needed to address the reviewers concerns takes about two months or less. If it will significantly longer than this, the policy is to reject the paper even if the potential is there for publication. Upon reconsideration, the original opinion about "too many problems" with "not enough time" holds; hence the rejected appeal. However, if you can address all the major concerns, *eLife* would reconsider a new submission on this topic. Technically this would be initially considered a new manuscript, but it would endeavor to have it considered by as many of the original reviewers as possible.

[Editors’ note: what now follows is the decision letter after the authors submitted for further consideration.]

Thank you for resubmitting your work entitled "High levels of histones promote Whole-Genome-Duplications and trigger a Swe1^WEE1^ dependent phosphorylation of Cdc28^CDK1^" for further consideration at *eLife*. Your revised article has been favorably evaluated by Kevin Struhl (Senior editor), a Reviewing editor, and two reviewers.

The manuscript has been improved but there are some remaining issues that need to be addressed before acceptance, as outlined below:

Genome stability is critical for normal physiology of an organism. Genomic instability at the chromosome level results in aneuploidy, which can cause birth defects and cancer. As cancer cells typically have chromosome numbers larger than the normal 46, one popular theory is that aneuploidy follows whole-genome duplication and polyploidy. Polyploidy puts more burden on the mitotic chromosome segregation machinery, increases errors in that process, and allows cells to better tolerate the resulting aneuploidy. Thus, it is important to study the potential causes and mechanisms of polyploidy.

Using the budding yeast as a model organism, Geli et al. showed in the current study that histone overexpression (coupled with or without mutations in other histone-surveillance genes) causes polyploidy. They further probed the underlying mechanisms and made two interesting findings. First, histone overexpression limits the deposition of the histone variant H2A.Z and its downstream effector condensin on chromosomes, particularly near centromeres. They believe that this mechanism underlies polyploidization caused by histone overexpression. Published results in the mouse and fly also link condensin to ploidy regulation, suggesting that the connection between condensin defects and polyploidization might be evolutionarily conserved. Second, histone overexpression stabilizes Wee1, promotes Wee1-dependent negative phosphorylation on Cdk1 and delays mitotic progression. Furthermore, histone overexpression activates the spindle checkpoint. These mechanisms are not, however, required for polyploidization, but actually impede it.

Overall, this study advances our understanding of ploidy regulation. The findings are novel and significant. The results are for the most part convincing and logically presented. Publication is recommended, provided that the authors address the following point.

Major point

1) The authors argue that histone overexpression activates the spindle checkpoint independently of the kinetochore. The major evidence supporting this claim is the lack of strong Mad2 foci formation. However, data in Figure 7 does show an increase of Mad2 foci formation. It thus remains possible that Mad2 is activated through the canonical kinetochore pathway. The authors should include positive controls in Figure 6 and Figure 7, such as cells treated with nocodazole. If the authors can't do additional experiments, then they should revise the text and discussion to reflect this caveat.

---

## [Author Response]

Reviewer #1:Miles et al. present multiple lines of evidence that histone overexpression causes mitotic defects in yeast cells. The DNA damage and spindle assembly checkpoints are not activated by such a stress, whereas Tyr_15_ of Cdk1 is phosphorylated, suggesting a novel Cdk1-dependent, but DDR- and SAC-independent regulatory function of mitosis. Whole genome duplication is an important phenomenon associated with development of many species and of human cancers. This work is thus of strong interest to many. However, this manuscript suffers from a slew of deficiencies, ranging from writing to experimental design to result interpretation. It'll take a very significant effort to resolve these issues satisfactorily.1) One of the most interesting and valuable observations from this work is probably the visualization of how diploids might be generated in the wake of histone overdose. That is, the movement of both sets of duplicated genome to the daughter cell without cytokinesis. However, the authors fall short on testing the functionality of other key mitotic machinery such as spindle, spindle pole bodies, separation of centromeres, etc. of these cells. Moreover, there is no telling whether these cells can proceed successfully to the next round of mitosis, resulting in the formation of a homogenous diploid population.

We are more than willing to try to perform live microscopy in the strain used in Figure 2 (that contains a tag to follow kinetochores, tubulin and the SPB) in a wild type strain carrying an empty vector or the 2μΔNEG construct. This experiment is feasible and within our capabilities but probably complex to analyze. Regarding the second question, we can try longer films and check if are able to catch a second division in cells that have experienced a WGD event. However, the facts that the observation of a WGD event is something that takes place at a low frequency added to the long time required for the cell to resume mitosis in the presence of histone overexpression may make this task rather difficult.

2) It is good that the authors use different approaches to induce histone stress, however, the basis of which remains unclear.

During this study we have tried several strategies to try to increase the amount of histones in the cell. There are two main reasons that lead us to finally decide to carry out our experiments with the HTA1-HTB1ΔNEG system. The first one is that this system allows us to overexpress histones when histones are not normally required (beyond replication) but should not interfere with the normal supply of histones required for DNA replication. The second one is that this construct was the only one among those tested (including all the plasmids in which histones are overexpressed using galactose inducible promoters) in which most of the phenotypes observed in the *rad53* and *tom1* mutants were reproduced in wild type cells. Moreover, cells -in which histone overexpression is driven by the GAL promoter- are quite difficult to release from G1 arrest. The reason to use a conditional strain (*rad53-AID lsm1Δ)* able to cut off all histone degradation is explained in the text (Results section). We are aware of the fact that *lsm1* and *rad53* play key roles in other processes in the cell but to our knowledge there is no direct evidence of a common function of these two proteins other than the regulation of histone levels during the cell cycle.

The authors cite a previous finding that Swe1 contacts H2B, but do not elaborate.

We do. Please check the second paragraph of the Discussion section.

They also suggest that reduced H2A.Z recruitment to chromatin might be responsible. However, there is no report of a similar mitotic defect in known htz1 alleles.

Chromosome segregation defects have been associated with *htz1* mutants (Keogh et al. Genes and Dev. 2006; Krogan et al. PNAs 2004) and polyploidy with H2A.Z overexpression (Chambers et al. Genes and Dev 2012). We agree that there is no direct link between the absence of H2A.Z and the generation of polyploid cells but our results do indicate that this phenomenon takes place at a low frequency and could have therefore passed unnoticed. For instance, polyploidy induced by telomere dysfunction was described just a few years ago (Davoli and de Lange, Cell 2010) despite numerous studies in the field of telomere.

It is also unclear how skewing histone expression would lead to WGD. For example, what is the minimal degree of overexpression needed? Figure 1 shows that GAL-controlled expression of all four core histones in the same rad53K227A background results in the retention of 1n population, suggesting that concerted expression of histones, not merely the number of histone molecules, is critical for mitotic integrity.

We agree, overall if we consider that Meeks-Wagner and Hartwell (Cell 1986) observed differences in the frequency of chromosome loss when histones were overexpressed in pairs (H2A-H2B or H3-H4) or simultaneously. This said, our results do allow us to conclude that both pairs of core histones expressed alone or in combination are sufficient to promote WGDs in a strain deficient for histone degradation. H2A-H2B overexpression is able to generate this phenotype even in wild type cells.

Furthermore, the authors favor the idea of downregulation of H2A.Z loading to pericentromeres being the underpinning of the observed mitotic defects. If so, will co-expressing H2A.Z with H2A and H2B alleviate at least some of the stress? How is it that H3/H4 overexpression also confers the growth phenotype as does H2A/H2B ectopic expression?

We were also initially puzzled by this result and propose a possible explanation based on our results in the Discussion section.

3) The ChIP-qPCR data are weak. What is the resolution of the qPCR? What are the reaction conditions? The provision of oligo primers in Table 3 does not help much.

We disagree. All our ChIP-qPCR experiments include a control sample (no antibody or no tag depending on the experiment) that has a signal at least one order of magnitude lower than the signal obtained for the proteins analyzed and is usually included in all figures. All comparisons mentioned in the text as significant normally include a paired t-test that is something not always present in many recently published papers. Regarding the resolution, we include an image at the end of this paragraph in which we can observe chromatin fragmentation from the first biological replicate. The DNA smear in this biological replicate is similar for all of them and seems to be more enriched in a region slightly smaller than 200 bp (last band of the ladder). We also include a first test that was performed with the *rad53-AID CSE4-MYC* (CENP-A) strain to determine the efficiency of the H2A.Z antibody and the MYC antibody (CENP-A). In this test, we can see that CENP-A is specifically enriched at both centromeres (CEN4 and CEN12) and that this enrichment is significantly decreased when the PCR set of primers is displaced less than 200 bp to the left or the right side of centromeres (CEN4 or 12 left and right). This test also shows that H2A.Z is highly enriched at intergenic regions and pericentromeric chromatin, two regions in which H2A.Z is normally present. Each primer used for the qPCR reactions includes a number with its exact position on the chromosome (start or end of the amplicon depending if it’s the forward or reverse primer). We can include a supplemental figure that maps the position of all set of primers.

4) Figure 7 MNase assays suggest that the subject chromatin is more resistant to MNase, indicating higher nucleosome occupancy. However, this single-dose experiment does not afford sufficient quantitative power. There are many advanced, more sophisticated approaches available for MNase assays. The authors should go deeper than the current endeavor.

We agree and we are willing to perform Mnase-Seq with the conditional strains if the paper is considered for review.

5) The experimental procedures provided are far from complete. Key experiments such as MNase digestion, cell lysate preparation for immunochemical analysis, and ChIP-qPCR are all left out.

We will include a detailed protocol for each of these techniques in the revised version if the paper is further considered.

6) There are many writing and grammatical errors throughout the text.

We sincerely apologize for this. We will pay more attention in the final version.

Reviewer #2:Genome stability is critical for normal physiology of an organism. Genomic instability at the chromosome level results in aneuploidy, which can cause birth defects and cancer. As cancer cells typically gain copies of chromosomes, one popular theory is that aneuploidy follows whole-genome duplication and polyploidy. Polyploidy puts more burden on the mitotic chromosome segregation machinery, increases errors in that process, and allows cells to better tolerate the resulting aneuploidy. Thus, it is important for us to understand the potential causes and mechanisms of polyploidy. Using the budding yeast as a model organism, Miles et al. showed in the current study that histone overexpression (coupled with or without mutations in other histone-surveillance genes) causes polyploidy. They further probed the underlying mechanisms and made two interesting findings. First, histone overexpression promotes Wee1-dependent negative phosphorylation on Cdk1 and delays mitotic entry. This mechanism is not, however, required for polyploidization, but may impede it. Second, histone overexpression limits the deposition of the histone variant H2A.Z and its downstream effector condensin on chromosomes, particularly near centromeres. They believe that this mechanism underlies polyploidization caused by histone overexpression.Overall, this study advances our understanding of ploidy regulation. The findings are novel and significant. The results are for the most part convincing and logically presented. Publication is recommended, provided that the authors address the following major points.1) While it is clear that H2A.Z and condensin deposition is defective under conditions of histone overexpression, the evidence linking this effect to polyploidization needs to be further strengthened. The only strong evidence presented is that deletion of the H2A.Z depositor Swr1 exacerbates the effect of histone overexpression. A genetic suppression experiment will be much more convincing. The authors should test whether H2A.Z overexpression or forced targeting of condensin to peri-centromeres reduces polyploidy caused by histone overexpression.

We are more than willing to test if H2A.Z overexpression is able to suppress WGDs in wild type cells transformed with the 2μΔNEG vector. The second experiment may be more complex to achieve. Indeed artificial targeting of condensin to pericentromeres may be detrimental for the cells.

2) Much of the results on DNA damage checkpoint, spindle assembly checkpoint, and Shugoshin are negative results. These sections should be shortened. This will leave more space for the authors to address point 1 above.

We can easily re-adapt the paper once all the experiments are finished in order to clarify the message.

Reviewer #3:This study builds on the notion that histone protein levels are tightly regulated by multiple mechanisms and that misregulation or histone overexpression causes toxicity, cell cycle arrest, sensitivity to DNA damaging agents, and mitotic chromosome loss.One key observation in this study is that conditions that cause increased histone protein expression lead to whole genome duplication events in yeast, i.e. cells fail to divide the nuclear material between mother and daughter cell in a timely manner; these cells stop proliferating, or the resulting diploid cells as a result of endoreduplication have a growth advantage and rapidly take over the population. This is also apparent on plates where diploids form larger colonies. Upon presumed severe increased histone levels (by disrupting the histone RNA and protein degradation pathways, which is lethal) cells arrest in G2/M and fail to exit from a G2/M arrest. In this first part, the authors investigated whether there is a sensing mechanism, or a checkpoint, for high histone levels. The failure to exit from G2/M was accompanied by a delay in cohesin degradation and increased Cdc28 phosphorylation by Swe1. The authors propose that this event delays the entry into mitosis. However, deletion of Swe1 abolishes Cdc28 phosphorylation but does not suppress the G2/M arrest. Swe1 loss does enhance WGD events.In the second part, investigating the cause of the WGD events, the authors show that presumed very high histone levels do not affect CENP-A recruitment to centromeres, but do impair H2A.Z and condensin recruitment to pericentric heterochromatin. A condensin mutation enhances the toxicity of histone overexpression and loss of the H2A.Z assembly factor Swr1 enhances the selection for diploid cells upon transformation with a histone H2A/B overexpression vector, together suggesting that loss of these two chromatin components can further exacerbate the effects of increased histone dosage. This leads the authors to suggest that the decrease in H2A.Z and condensin is at least one of the causes that trigger WGDs when excess free histones accumulate. However, another interpretation of these results (in essence negative genetic interactions) is that histone excess and loss of H2A.Z/condensin act by independent mechanisms because a loss of H2A.Z alone was not sufficient to cause WGD events. It is likely that increased histone dosage affects WGD also by other mechanisms.This study presents a series of interesting observations that warrant further investigation. However, the claims made by the authors are not always justified and supported by data. Importantly, whether there is a CDC28/Swe1-mediated sensing mechanism has not been fully addressed. In addition, for many of the experiments, replicates are missing and the study would benefit from a more quantitative appreciation of the altered histone levels. Finally, the story is composed of two pieces that are currently not well connected.- The claims made by the authors are not always justified and supported by data, e.g. "we have uncovered a new type of stress (histone-stress) able to trigger a stabilization of Swe1^WEE1^ that promotes Cdc28^CDK1^ phosphorylation and delays the entry into mitosis". Similar conclusions are drawn elsewhere (e.g. Discussion, third paragraph and Abstract). That histone overexpression causes stress was already known. That it stabilizes Swe1 is not shown in this manuscript. Furthermore, that Cdc28 phosphorylation delays entry into mitosis is not so clear because deletion of Swe1, the responsible kinase, does not affect the delay.

Of course, we are aware that we are not the first ones reporting that large accumulation of histones causes stress in cells or the first ones that demonstrate that Swe1 is a stress responsive kinase able. What we want to point out with this sentence is that we have uncovered a type of stress able to trigger a Cdc28^CDK1^ phosphorylation that depends on the stress-responsive kinase Swe1^WEE1^. We agree that Swe1 stability is not directly addressed in the paper. However, we do show that Cdc28 phosphorylation in *rad53-AID lsm1Δ* treated cells depends on the presence of this kinase and does not take place when Swe1 is lacking. We have recently found a specific antibody for Swe1 that we could order and use it to address this issue in Figure 5. The conclusion of the delay is based on the results obtained in Figure 5 and not in Figure 6 (this result is discussed in the second paragraph of the Discussion section). In Figure 5 we do observe that *rad53* cells increase the levels of phosphorylated Cdc28 and the time required to reach the next G1 (compare the FACS profile at 105 and 120 minutes) specifically when these cells are exposed to high levels of histones H2A and H2B.

- For many key experiments replicates or quantification of replicate experiments are missing. For most of the western blots, FACS analyses, microscopy, and MNase sensitivity assays in the manuscript no replicates or quantification of replicates are provided. The authors should provide additional data on these experiments to demonstrate reproducibility, following the guidelines of eLife.

We would like to apologize for this issue. The new revised version will include an additional source data file that contains all the raw and processed data and a pdf file with original images for all the replicates performed in all key experiments. We are also happy to share raw data from live microscopy videos if there is a server in which we can upload them.

- The authors use various conditions to increase histone expression outside S-phase. One approach is overexpressing the histone genes from strong promoters. This is in principle straightforward. The other main approach is combining a Rad53 degron with an lsm1 mutation, which causes cell death. There are several issues with the approaches.1) In the latter case (lsm1+rad53-aid) it cannot be excluded that the phenotypes seen are at least in part caused by other functions of the inactivated proteins. Although this is partially addressed in Figure 1, the Cdc28 phenotype has only been observed in rad53 mutant backgrounds.

Please check the answer to reviewer 1 (Point 2 major issues).

2) The authors use various conditions to induce histone overexpression but show in very few cases to what extent histone proteins are overexpressed. This is an important issue because the various methods and combinations of alleles and/or plasmids have different phenotypic consequences (e.g. 2-micron-δ NEG vs. lsm1+rad53-aid). The authors should examine histone protein levels for the different conditions used, for example not only lsm1 with or without rad53-aid (Figure 3), but also for the conditions used in Figure 1 and Figure 5. For example, based on the phenotypes, one might assume that the δ NEG or GAL-H2A-H2B plasmids cause lower histone protein expression than the combined rad53+lsm1 mutation. However, this is not demonstrated by histone blots.

We have now examined histone levels in the 2μΔNEG vector by RT-PCR and western blot. We will include this data in the revised version.

3) The assay first shown in Figure 2 involves transformation with a 2-micron-HTA-HTB1-δ-NEG plasmid. The authors monitor diploidization of the pool of cells over time starting at the time of transformation. This is a somewhat unconventional assay that would benefit from more explanation of the approach and experimental details. For example, at what time point was the vector control analyzed? It would be appropriate to take a vector control along for every time point and also take a t=0 point. In this assay several factors can influence the outcome or dynamics: First, after transformation only a small proportion of the cells will contain a plasmid and survive the selection, most cells will die. Therefore, the kinetics might depend on the transformation efficiency. Second, cells that become diploid will have a growth advantage and take over the population. How fast they do so will depend on the rate of diploidization as well as the growth rate. These different parameters should be explained more clearly and taken into account experimentally. Another point here concerns panel 2D: what is the time point after transformation, how did the authors assess whether the cells examined actually contained a histone plasmid, and what are the numbers for a wild-type cell or vector control?

All experiments involving the *ΔNEG* cassette started from exponentially growing cells transformed with a centromeric (*rad53K227A* and *tom1Δ)* or a 2-μ vector (wild type) carrying the *HTA1-HTB1ΔNEG* construct. After transformation, strains were plated on SC-Uracil selective media at 30ºC. Time courses to determine the timing of WGDs were performed always from single small colonies coming usually from a 5-days growth plate. These cultures were grown and maintained in exponential growth conditions in liquid SC-uracil minimal media and were only considered for further analysis when they had a full haploid DNA content the first day (day 2 or 1) of the kinetic. Number of days in these experiments reflects the number of days that cells were grown in liquid media culture and not the number of days after transformation. Each experiment performed carried at least one colony of the strain tested transformed with an empty vector. This control cell was treated essentially as cells transformed with the histone-overexpressing vector. Samples were taken daily to estimate the proportion of haploids and diploids by FACS.

Other Points:Figure 1. How long were the cells grown in galactose to induce histone expression?

They were maintained in galactose before and after transformation.

The authors state that 'Rad53 depletion was lethal in the absence of Lsm1 and largely increased the amount of histones in the cell (Figure 3).' However, the increase in H2B seems modest.

We agree but it is something that seems to be quite repetitive in our four biological replicates. It is important to consider that histones are very abundant proteins in the cell and that even a small increase can represent a very large amount of additional histone molecules in the cell.

In Figure 3, the colony formation data should be represented as absolute numbers instead of relative

We can change this graph by a new one that represents all the different cell types observed for each mutant and each condition.

In Figure 5, the blots are confusing. In the top panel it is hard to judge the absence of the band in the last lane, in the bottom panel it is not clear why the intensity of the two bands changes?

We agree and we are willing to repeat this experiment to obtain images with a better quality.

In Figure 5, the cell cycle progression is hard to judge from the images shown; the phenotype of the rad53 mutant seems independent of the histone plasmid (and this may also be the case for cdc28 phosphorylation).

We disagree. The FACS profiles and the levels of Cdc28 phosphorylation are clearly different at time 105 and 120 in the *rad53* mutant when histones are overexpressed.

Furthermore, in the third paragraph of the subsection “Histone accumulation triggers a Cdc28^CDK1^ phosphorylation that delays the entry into mitosis” the authors conclude that the phosphorylation in the upper right panel was slightly increased. However, this cannot be concluded from the blots shown because the comparison is made between two separate blots.

We agree and will eliminate this comment.

For Figure 7–Figure 8 it is not clear how many biological replicates were performed.

All experiments were at least three times (unless indicated in the figure legend) using three independent biological replicates. We can correct this in the figure legend.

The authors suggest that 'rad53 AID lsm1Δ treated cells were more resistant to MNase digestion' (Figure 7, and subsection “High levels of histones decrease Htz1^H2A.Z^ and condensin incorporation to pericentromeric chromatin”, second paragraph). This is not immediately obvious from the figure. The authors should provide additional analyses or discussion to support this conclusion.

We agree and we are willing to perform Mnase-Seq with the conditional strains if the paper is considered for review.

Regarding the mechanism of WGD by increased histone dosage, a role for H2A.Z in condensin stability has previously been shown in S. pombe. Given the differences in pericentric chromatin between the models yeasts, it would be useful to establish whether there is also a connection in S. cerevisiae.

H2A.Z and condensin recruitment to pericentromeric chromatin is conserved from yeasts (*S.cerevisiae* and *S.pombe)* to higher eukaryotes (mouse and humans). Both of them play an important role during chromosome segregation and lead to chromosome segregation defects when they are absent. There is no direct evidence to our knowledge of a physical interaction between H2A.Z and condensin in *S. cerevisiae*.

[Editors' notes: What follows is the response to the previous round of reviews included in the authors' resubmission]

We are sending you a second revised version of our manuscript now entitled “High levels of histones promote Whole-Genome-Duplications and trigger a Swe1^WEE1^ dependent phosphorylation of Cdc28^CDK1”^. The previous version was entitled “A histone-sensing response prevents undesired endomitosis promoted by histone overexpression”. We have changed the title according to a comment raised by one of the reviewers.

In our previous submission, you explained us that all three reviewers thought that the work was of sufficient interest for *eLife* and that they would be willing to consider a new/revised submission that addresses all of the issues raised by the reviewers. You mentioned “Technically this would be initially considered a new manuscript, but we would endeavor to have it considered by as many of the original reviewers as possible”.

Reviewer #1:The authors are praised for their endeavors to probe from different angles for the very interesting histone overdose-related cell division phenotypes in the budding yeast. The discoveries of the potential involvement of H2A.Z and condensin are very promising.However, there are insufficiencies need to be addressed.a) One of the fundamental issues that I have for this manuscript is the use of "endomitosis" for the observed phenotype. There are well-documented and -defined examples, normal or pathological, for endomitosis and endoreplication. The authors' initial observation might be better described as an escape from failed segregation, so that a small number of daughter cells inherit the full set of duplicated genomes, at the cost of losing the mother cell. Oddly, the "diploidization" stops at a 2n DNA content, and does not progress into 4n, 8n, or even higher ploidy that is the case for typical endocycles.

The term endomitosis was used to describe the phenomenon observed in Figure 2 in which the nucleus divides inside the daughter cell without cytokinesis. To avoid confusion, we have replaced the term “aberrant endomitosis” by “aberrant cell division”. We have detected 4n cells in rare occasions while we were analyzing the DNA content of rad53K227A and tom1∆ cells transformed with the CEN∆NEG vector.

We have never observed them in wild type cells transformed with the 2µ∆NEG vector. We think that this issue could be related to the fact that a 2N DNA content in wild type cells is sufficient to buffer the toxic effect generated by high levels of histones in haploid cells.

b) Another concern is that overexpressing histones H2A and H2B is sufficient to trigger the 2n phenotype in a wildtype background. The use of the rad53-AID lsm1 null strain does not add significant values to the study but might further complicate the matter (e.g., rad53 loss-of-function alleles can cause problems in cell cycle progression). Therefore, mechanistic in sights from the use of these strains may not be directly applicable to WGD caused by histone overexpression.

Rad53 and Lsm1 have no other function in common than the regulation of histone levels to our knowledge. However, it is certainly important to consider the possibility that some of the phenotypes observed in the double mutant may be indirect and not related to histone metabolism. To prove if the defect in H2A.Z incorporation observed in the double mutant was relevant for WGDs we have measured H2A.Z levels, and checked the effects of H2A.Z overexpression and SWR1 deletion in cells transformed with the 2µ∆NEG (new Figure 5). All the results are consistent with the hypothesis that H2A/H2A.Z balance is affected by histone overexpression and critical in the generation of WGDs.

To prove that Cdc28 phosphorylation is also directly related to histone levels, we have followed Cdc28 phosphorylation during a whole cell cycle in cells that were overexpressing histones H2A and H2B in a wild type and in rad53K227A cells in which histones are not efficiently degraded after histone overexpression (Gunjan et al., Cell 2003). Our results in the rad53K227A indicate that Cdc28 phosphorylation is maintained for a longer time in cells in which histones are being overexpressed.

c) Technical issues need to be addressed as well. Firstly, the characterization of histone dosages and chromatin distribution is less than ideal. The authors cherry-picked the methods and histones for analysis. Why not just purify all core histones and resolve them by SDS-PAGE for CBR staining?

As asked by reviewer 1, we have purified histones (Jourquin and Géli, 2017; Histones Methods and Protocols) in rad53-AID and rad53-AID lsm1∆ mutants treated or not with Auxin and replaced the results from the previous version (Western Blot and Chromatin Fractionation). Histones were resolved by SDS-PAGE and stained with CBR. Histone amounts were normalized to a non-specific band present in the purification. We observe a relative increase of histones compared to a non-specific band in the rad53-AID lsm1∆ double mutant treated with Auxin either compared to the non treated sample or to the rad53-AID single mutant treated or not (New Figure 3).

d) The authors argued that H2A.Z is the key player for WGDs, and yet the characterization of H2A, the counterpart of H2A.Z, is lacking (except Figure 8). Did they see the expected inverse correlation between H2A and H2A.Z?

As asked by reviewer 1, in the revised version of the manuscript we have performed ChIPqPCR with an H2A antibody (new Figure 5). We observe that the level of H2A increases at those regions exhibiting a decrease of H2A.Z (new Figure 5).

e) Secondly, Figure 7 shows that the centromeric Cse4 level does not change, yet histone H3 detection at centromeres shoots up by a factor of about 10, and H4 level is up at the similar level at centromeres. What is the explanation? Why does H3 not evict Cse4?

*S. cerevisiae* has just one centromeric nucleosome able to incorporate either one (Dalal et al., Plos Biology 2007) or two Cse4 molecule/s (Mizuguchi et al., Cell 2007; Wisniewski et al., *eLife* 2014). Our Cse4 ChIP indicates that the amount of Cse4 at centromeres does not change when histones are overexpressed (Figure 4) and therefore disfavors a model in which H3 competes with Cse4 incorporation. The fact that Mtw1, a subunit of the MIND complex required to build a functional kinetochore, is recruited with a similar efficiency and always remains in line with tubulin filaments (Figure 4), further suggests the formation of a functional kinetochore that is able to attach chromosomes to microtubules. We agree however with reviewer one that the large increase in H3 and H4 at centromeres and other regions of the genome is intriguing. We discussed thoroughly this point and propose two different explanations.

The first one is that histone overexpression favors the incorporation of additional nucleosomes at both sides of the centromeric nucleosome. The chromatin that surrounds the centromere is far from perfectly positioned (Cole et al., PNAS 2011) and may allow the incorporation of additional nucleosomes that will increase the signal at centromeres. This hypothesis is simple but has to face the fact that there is a space constraint. Indeed, only a limited amount of nucleosomes can be incorporated in 200600 bp fragments, the average size of DNA fragments that we obtain after sonication in our ChIP experiments. Therefore, the large increase in histone ChIP levels that we observe would be difficult to explain with the addition of 1 or 2 nucleosomes.

The second explanation that could explain this increase would be a change in chromatin compaction. Nucleosomes can physically interact with neighbor nucleosomes and promote chromatin condensation (Wilkins et al., Science 2014). Changes in chromatin condensation allow the interaction of nucleosomes with other nucleosomes that are not necessarily assembled at the same region. These nucleosomes should, in principle, retain these interactions after formaldehyde crosslinking, and should be able to increase the ChIP signal even if these nucleosomes are not directly interacting with centromeric chromatin.

f) Thirdly, overexpressing H2A.Z reduces the percentage of 2n* cells. Is this genuine suppression or that cells suffer from lower viability, hence the percentage of viable 2n* colonies decreases? Are the overall chromatin structure and condensin localization improved?

As demanded by reviewer 1, in the revised version we show the growth of haploids and 2n* cells transformed with the vector allowing the overexpression of histone H2A.Z or an empty control vector (Figure 5—figure supplement 1). We observe no obvious differences in growth between n and 2n* cells overexpressing H2A.Z or carrying the empty vector control. In cells transformed with the 2µ∆NEG vector, H2A.Z overexpression enhances the growth of both haploids and 2n* cells. We do not observe a counter-selection of diploids.

We have also performed ChIP experiments to see if H2A.Z overexpression is able to suppress the defect in the incorporation of H2A.Z observed in rad53-AID lsm1∆ treated cells. We observe that H2A.Z overexpression is not sufficient to restore the levels of H2A.Z at pericentromeric chromatin in this double mutant. The rad53-AID lsm1∆ double mutant must have other defects that are able to maintain lower levels of H2A.Z incorporation to chromatin. We propose a possible explanation in the second paragraph of the Discussion.

g) Lastly, with all the supplemental figures, why did authors choose to not show the mating characterization results of the 2n* cells?

We did not include it because we considered it something anecdotic and not really important for the overall message of the paper. As asked by reviewer 1, we have included it in the new version (Figure 1—figure supplement 1).

Reviewer #2:This manuscript appears to be the re-submission of a previously submitted manuscript. I have attached my original review along with comments on the newly added data.Genome stability is critical for normal physiology of an organism. Genomic instability at the chromosome level results in aneuploidy, which can cause birth defects and cancer. As cancer cells typically gain copies of chromosomes, one popular theory is that aneuploidy follows whole-genome duplication and polyploidy. Polyploidy puts more burden on the mitotic chromosome segregation machinery, increases errors in that process, and allows cells to better tolerate the resulting aneuploidy. Thus, it is important for us to understand the potential causes and mechanisms of polyploidy. Using the budding yeast as a model organism, Miles et al. showed in the current study that histone overexpression (coupled with or without mutations in other histone-surveillance genes) causes polyploidy. They further probed the underlying mechanisms and made two interesting findings. First, histone overexpression promotes Wee1-dependent negative phosphorylation on Cdk1 and delays mitotic entry. This mechanism is not, however, required for polyploidization, but may impede it. Second, histone overexpression limits the deposition of the histone variant H2A.Z and its downstream effector condensin on chromosomes, particularly near centromeres. They believe that this mechanism underlies polyploidization caused by histone overexpression.Overall, this study advances our understanding of ploidy regulation. The findings are novel and significant. The results are for the most part convincing and logically presented. During the review of the initial draft of this paper, I raised the point that the connection between polyploidization and H2A.Z/condensin was weak, and asked for genetic suppression data. The authors have now provided convincing evidence to show that H2A.Z overexpression reduces the extent of polyploidization caused by canonical histone overexpression. This new piece of evidence strengthens their argument for a link between H2A.Z and polyploidization. In a second development, I recently heard at a meeting that condensin II mutations cause tissue-specific polyploidization in the mouse. The finding was actually published near the end of 2016 (Woodward et al. (2016) Genes Dev, 30:2173-2186), but apparently escaped my notice (and was not referenced in this paper). In the Woodward et al. study, Andrew Wood and coworkers showed that mutation of condensin II caused lymphoma and polyploidizition of tumorinitiating cells in the mouse. These findings, along with results in the fly (which the authors discuss), I am now convinced that there is an evolutionarily conserved connection between condensin defects and polyploidization. For these reasons, I recommend the publication of this excellent study.

We really appreciate the comments from reviewer 2. We have included this reference in the new version.

Reviewer #3:This study presents several interesting findings. First, increased histone dosage in budding yeast leads to genome segregation errors and whole-genome duplication. Second, increased histone dosage triggers a series of events that can be described as a sensing mechanism. However, the function of this sensing mechanism in relation to WGDs remains elusive. The sensing mechanism seems to be independent of known checkpoint mechanisms. Finally, increased histone dosage leads to reduced H2A.Z levels in the cell and reduced condensin loading in pericentric regions; these events may contribute to the genome instability caused by histone stress. Although many questions remain to be answered, together, the results of this study will be of interest to a broad audience.In this revised and improved version of the previously submitted manuscript the authors now show histone expression levels for some of the experiments and show Swe1 expression data to confirm its stabilization. In addition, more information is provided about the replicates and experimental conditions. Finally, the manuscript has been rewritten and generally, the claims made by the authors have a better match with the results than in the previous version. However, several concerns remain.In the summary, the statement 'uncovers a mechanism able to sense histone levels before mitosis in order to avoid their undesired consequences' is not fully supported by the data. The function of the sensing mechanism still remains unclear.In the Discussion, it is mentioned that 'Swe1 delays mitosis in the presence of high levels of histones'. The authors do not provide evidence to support this claim.

We have rewritten the sections related to the role of checkpoints to make a new version that is more descriptive and less conclusive. The statement that Swe1 is able to sense histone levels before mitosis in order to avoid their undesired consequences has been tuned down in the Summary, Results and Discussion. Possible consequences of Cdc28 phosphorylation on the cell cycle are mentioned in the discussion.

In Figure 5, the authors show Swe1 blots to support the notion that Swe1 is stabilized upon histone stress. However, a loading control is missing. Cdc28 blots are shown but this protein is related to Swe1 function and it changes due to phosphorylation. Panel A also misses a loading control.

We have included Act1 as a loading control in the three figures, which are now shown in the new Figure 6) (previous Figure 5). See also our response to referee 1 (point 6). Cdc28 levels remain constant throughout the cell cycle in *S. cerevisiae* (Mendenhall and Hodge, 1998).

In Figure 8, the authors show on blots that H2A.Z and H2.A levels change in opposite directions upon histone stress. However, why is the level of H2A.Z already low in rad53-AID lsm1 without treatment? Similarly, why are H4 levels lower in rad53-Aid treated vs untreated?

We believe that they represent loading differences rather than real differences in the levels of H2A.Z. In the new version of the manuscript, western blots are performed with histonepurified extracts. In the new blots performed with these samples (New Figure 3) this difference is no longer observed.

The authors now explain more clearly in the text on page 7 how the assay for Figure 2 (and subsequent figures) was done: Cells that remained completely haploid were inoculated from plates into liquid media and then monitored over time. Here it would be appropriate to show that at t=0, the cells indeed started off as haploids. The vector control does not provide this information. And what is time point used for the vector control, why is only one time point shown for the vector alone? What was the time point used for Figure 2?In this study, FACS profiles are always made from exponentially growing cells in liquid cultures and not from cells directly collected from petri dishes. This decision was based on the fact that plated cells after 4 or 5 days tend to accumulate in G1/G0. As a result, diploid populations are difficult to differentiate from haploid cells at a G2/M stage. Obtaining a sample in exponential growth conditions usually takes two days in cells transformed with the 2µ∆NEG vector due to their slow growth. Day 2 therefore constitutes the first time-point for which we can harvest enough cells to perform a FACS experiment. At day 2, a large proportion of wild type cells transformed with the 2µ∆NEG vector remain haploid. However, in some cases, some of the transformed colonies have already started to change to a mixed population of haploids and diploids.

The vector control is used to prove that the strain is completely haploid before transformation with the 2µ∆NEG vector and also to prove that this strain does not go to a 2n state in the absence of the 2µ∆NEG vector. We have always included at least one in each kinetic and taken samples the first and the last day. The one shown in Figure 2 corresponds to the one taken the last day. We have never observed a diploidisation in any genetic context in the absence of histone overexpression. That is why we only represent one sample and not two. We believe that is important to point out that *S. cerevisiae* cells are very stable unicellular organisms that do not usually experience changes in their ploidy content after subcultivation.

In Figure 1—figure supplement 1, the authors show doubling time data for haploids transformed with the multi-copy deltaNEG plasmid. However, in Figure 6—figure supplement 1, the authors show that these cells rapidly diploidize. Please explain.

There is a certain variation in the time taken by each transformant to become fully diploid, something that can be appreciated in the kinetics shown in the new Figure 5 and Figure 8. Duplication times of transformants were estimated from a twelve-hour time course performed during day 2. FACS samples were taken at the beginning and at the end of the time course to check the ploidy. Samples that were already diploid or had started to diploidize during the kinetic were not considered for the estimation of the duplication time.

For many figures, the authors show the average of two experiments and a standard deviation. Although two replicates may sometimes suffice, including more replicates gives more confidence in the results. For small sample size, showing the two data points or the spread is more informative than standard deviation.

In this new version we have corrected the representation of our results according to the suggestion of reviewer 3. We have also extended the number of repeats of two of our key experiments, the condensin ChIP experiment, that now shows the average of five biological replicates and the suppression of WGDs by H2A.Z overexpression, that is done with 4 independent biological replicates. Most of the other results for which only two repeats are shown, are measured in more than one experiment (Mtw1 in the new Figure 4; Pds1 in the new Figure 6; Scc1 in the new Figure 6 and old Figure 4; Cdc28 phosphorylation in the new Figure 6), or demonstrated using different approaches (accumulation of histones in the rad53-AID lsm1∆ treated cells was shown by western blot and chromatin fractionation in the old version and is confirmed with histone purification in the new version and its incorporation to chromatin was shown with ChIP and Mnase).

For Figure 5—figure supplement 1, why was the plating done on Gal media, did these strains express an inducible copy of histone genes?

The AID system includes an adaptor protein required for the degradation that is under the control of a galactose inducible promoter. This information is included in the Supplemental Experimental Procedures.

Figure 6—figure supplement 1: it is not clear what the error bars refer to and what statistical test was performed.

It was included in the additional source data file but we forgot to include it in the figure legend. We have corrected this and included the statistical test in the new version.

In Figure 8 HTZ1 overexpression is used. Please provide the details of the HTZ1 vector and the vector control.

The details have been included in the new version.

In Figure 5, the blots are confusing. In the top panel it is hard to judge the absence of the band in the last lane, in the bottom panel it is not clear why the intensity of the two bands changes?

We have performed new blots that include Act1 as a loading control.

[Editors’ note: the author responses to the re-review follow.]

Major point1) The authors argue that histone overexpression activates the spindle checkpoint independently of the kinetochore. The major evidence supporting this claim is the lack of strong Mad2 foci formation. However, data in Figure 7 does show an increase of Mad2 foci formation. It thus remains possible that Mad2 is activated through the canonical kinetochore pathway. The authors should include positive controls in Figure 6 and Figure 7, such as cells treated with nocodazole. If the authors can't do additional experiments, then they should revise the text and discussion to reflect this caveat.

We have revised the text accordingly. We acknowledge the increase of Mad2 foci

formation shown in Figure 7:

“The fact that Mad2 foci were slightly increased (Figure 7) suggest that the SAC could be activated at least in some cells when histones are overexpressed.”

“The Pds1 stabilisation observed in *rad53*-AID *lsm1*Δ treated cells added to the fact that MAD2 deletion enhances WGDs indicate that the SAC could be activated in cells that overexpress histones.”